# TRANSFORMERS AS MULTI-TASK LEARNERS: DECOUPLING FEATURES IN HIDDEN MARKOV MODELS

## ABSTRACT

Transformer-based models have shown remarkable capabilities in sequence learning across a wide range of tasks, often performing well on specific task by leveraging input-output examples. Understanding the mechanisms by which these models capture and transfer information is important for driving model understanding progress, as well as guiding the design of more effective and efficient algorithms. However, despite their empirical success, a comprehensive theoretical understanding on it remains limited. In this work, we investigate the layerwise behavior of Transformers to uncover the mechanisms underlying their multi-task generalization ability. Taking explorations on a typical sequence model—Hidden Markov Models (HMMs), which are fundamental to many language tasks, we observe that: (i) lower layers of Transformers focus on extracting feature representations, primarily influenced by neighboring tokens; (ii) on the upper layers, features become decoupled, exhibiting a high degree of time disentanglement. Building on these empirical insights, we provide theoretical analysis for the expressiveness power of Transformers. Our explicit constructions align closely with empirical observations, providing theoretical support for the Transformer's effectiveness and efficiency on sequence learning across diverse tasks.

## 1 INTRODUCTION

Transformer-based models have achieved state-of-the-art performance across a broad range of sequence learning tasks, from language modeling and translation (Touvron et al., 2023; Dubey et al., 2024; Achiam et al., 2023; Team et al., 2023) to algorithmic reasoning (Liu et al., 2024; Ye et al., 2024). Remarkably, a single Transformer can often generalize across diverse tasks with minimal supervision, leveraging only a few input-output examples—a capability that underpins its success in few-shot and in-context learning (Brown et al., 2020; Wei et al., 2022; Dong et al., 2023; Min et al., 2022).

While the empirical success is well-documented, a key question remains elusive:

*How do Transformers capture and transfer information across layers?*

Understanding these internal mechanisms is crucial for advancing algorithmic design and developing more efficient model architectures. In particular, the internal mechanisms by which Transformers represent and process sequential information across layers are not yet fully understood. This gap is especially pressing given the growing interest in deploying large-scale Transformers in multi-task and general-purpose settings.

In this work, we aim to bridge this understanding gap by investigating the layerwise behavior of Transformers. We take explorations on Hidden Markov Model (HMMs)(Rabiner, 1989; Baum & Eagon, 1967), a classical class of sequence models where observations depend on unobserved hidden states evolving underlying Markov dynamics. Through empirical analysis, we uncover that while achieving good performance, Transformer learns feature representations on the lower layers, which are heavily influenced by nearby tokens, as well as developing decoupled features on upper layers, behaving like time disentangled representations (see Section 2 for details). Motivated by these observations, we provide a theoretical analysis of Transformer expressiveness. By constructing explicit Transformer architectures that model HMMs efficiently, we demonstrate how the observed

empirical patterns naturally emerge from our constructions. These results offer principled insights into how Transformers capture and generalize sequence information across tasks, shedding light on their success in multi-task and few-shot learning. Such feature decoupling phenomenon may also have potential practical implications, such as improving inference efficiency by design parallel computing on upper layers, which might be valuable future directions. Our main contributions are summarized as follows:

1. **Expressiveness.** On the theoretical side, we model language tasks in Transformers through Hidden Markov Models. Given the large hidden state space often encountered in practice, we adopt a low-rank structure for latent transitions, which has received tremendous attention recently for its efficiency in computation and inference (Siddiqi et al., 2010; Chiu et al., 2021). We show that under mild observability assumptions, Transformers can approximate low-rank HMMs using a fixed-length memory structure, enabling effective in-context learning. On the empirical side, we present that well-trained Transformers achieve high accuracy under in-context learning, with performance improving as more input-output examples are provided or as sequence length increases, which aligns with Theorem 1.

2. **Feature Decoupling Phenomenon.** On the empirical side, we observe that lower layers focus on learning local representations, primarily influenced by neighboring tokens. Upper layers develop decoupled, temporally disentangled representations that are less tied to specific input positions and encode higher-level abstractions. Our theoretical constructions provide corresponding explanations: lower layers extract local features, which are then transformed into decoupled, task-relevant representations in upper layers.

3. **Generalization to ambiguous settings.** We extend our theoretical results to more challenging scenarios where the hidden state space exceeds the observation space, which are natural assumptions in NLP. And we show that Transformers can still learn expressive representations by composing features from multiple future observations.

4. **Technical contribution.** From the technical level, we first provide a theoretical analysis of sample complexity on causal tasks, establishing a quantitative relationship between sample size, model capacity and prediction performance.

## 1.1 RELATED WORKS

The expressiveness of Transformers on sequence modeling has been explored from several perspectives. Liu et al. (2022a) demonstrate that Transformers can emulate automata by learning deterministic transition patterns. Nichani et al. (2024) analyze a simplified setting where the data follows a Markov chain governed by a transition matrix. Other works, such as Sander et al. (2024) and Wu et al. (2025), study the expressiveness of Transformers in autoregressive modeling, focusing on non-causal tasks. In contrast, our work takes a first step toward understanding the expressive power of Transformers on Hidden Markov Models, which are arguably among the simplest yet fundamental tools for modeling natural language tasks.

The detailed related works can be seen in Appendix A.

## 2 STARTING FROM THE EMPIRICAL FINDINGS

### 2.1 EXPERIMENT SETTINGS

To empirically investigate how Transformers learn multiple tasks on sequential data, we construct a dataset generated by a mixture of Hidden Markov Models (HMMs). Each HMM is used to model a tasks-specific distribution, and by mixing them we get a dataset similar to a pre-training corpus to learn language modeling on. In specific, we randomly simulate 8192 HMMs. The generation process is as follows. There is an initial task distribution on which we sample the HMM id. Each HMM composes of 128 hidden states

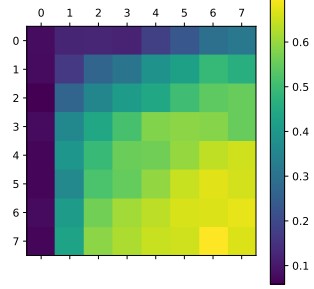

Figure 1: Accuracy of the Transformer under in-context learning setting. The y-axis denotes the number of demonstrative examples in-context, and the x-axis denotes the length of the test input $o_{test}$. All demonstrative examples have a length of 8 in this setting.

randomly transiting between each other. Each next state depends purely on the previous state, making the sequence of hidden states Markovian. All HMMs share a 16-token vocabulary. Each hidden state is associated with an emission distribution to randomly output a token. We sample 131k data, which allows training for 64 epochs[1], with 64 steps in each epoch on a batch size of 32. We build a transformer of 16 layers and 16 heads in each layer, and a hidden state dimension of 1024.(Verifications on other models are in Appendix B.) The transformer adopts the design of Roformer Su et al. (2024) which uses rotary positional encoding technique, and determines the attention logit between two tokens based on their relative position.

**Remark 1.** *Unlike Edelman et al. (2024); Park et al. (2024), our focus is not on task-level generalization to unseen HMMs, but on the model's ability to adapt to a new sequence realization and infer the latent dynamics in-context, which aligns with the definition used in GPT-3 (Brown et al., 2020) and many empirical ICL works.*

## 2.2 RESULTS

**Expressiveness power on HMMs.** Figure 1 iterates over per-sample length (x-axis, from 1 to 8) and the number of samples (y-axis, also 1 to 8), and reports the ICL accuracy obtained from these prompts. The high accuracy observed in Figure 1 highlights the expressiveness of well-trained Transformers. Moreover, we find that (1) accuracy improves as the number of input-output examples increases, and (2) task outputs become more predictable with longer test sequences.

**Decoupled features on upper layers.** Figure 2 is produced by shuffling the demonstration examples, and tracking how each sample's received logit change as the same sample is moved to different positions. The color shows the stability metric $1 - \text{std}/\text{mean}$. As each attention head assigns a logit distribution, we plot a matrix to illustrate each head (x-axis) and layer (y-axis). As shown in Figure 2, the upper layers (layers 9–15) exhibit attention logits that are less dependent on the positions of input tokens. This suggests that feature representations in these layers become increasingly decoupled, reflecting a high degree of time disentanglement.

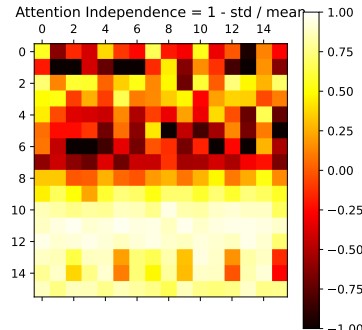

Figure 2: After randomly shuffling the positions of demonstrative inputs, we examine how the logits receive changes over layers (y-axis) and attention heads (x-axis). The measure is $1 - \frac{\text{std}(\text{logits})}{\text{mean}(\text{logits})}$.

**Layerwise investigations on Transformer recognitions.** Figure 3a and Figure 3b are generated by probing the intermediate layer representations with linear classifiers to test whether they contain task IDs, hidden-state IDs, or previous-token information. In the probing experiments, it is computed by training linear classifiers on the hidden representations from each layer and reporting standard classification accuracy. Figure 3a shows that Transformers gradually recognize the task identity across layers. Within a single task, the hidden state is identified earlier than the task itself, indicating that Transformers first learn the relationship between observations and hidden states in the lower layers, and then capture task-level structural information in the upper layers. This reflects a layerwise processing hierarchy in how Transformers handle sequential information. In Figure 3b, we observe three key patterns: (1) The Transformer identifies previous tokens ($i-1$, $i-2$, $i-3$, $i-5$, $i-10$) with decreasing accuracy as the distance increases, suggesting that feature learning in lower layers relies primarily on nearby tokens. (2) The accuracy curves for all distances follow a rising-then-falling trend across layers, implying that Transformers initially aggregate information from local contexts, and the resulting features then act as decoupled representations in upper layers. (3) The peak of each curve shifts to upper layers as the distance to the previous token increases, showing that Transformers first integrate information from close neighbors and then progressively attend to more distant tokens.

**Remark 2.** *In this work, our primary goal is to clarify and explain the feature-decoupling phenomenon, which we believe may have implications for practical large-scale models. For example, as*

---

[1]The term "epoch" in our implementation refers to a training cycle consisting of 64 gradient steps, each with a batch size of 32. This usage follows standard practice in large-scale language-model pretraining, where the data stream is effectively infinite and an "epoch" denotes a fixed number of optimization steps rather than a full pass over a dataset.

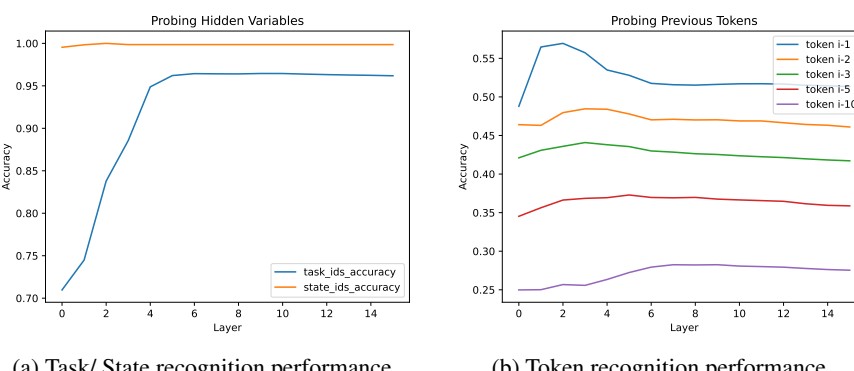

(a) Task/ State recognition performance.  (b) Token recognition performance.

Figure 3: Investigation on Transformer recognitions.

*features become decoupled in higher layers, it may be possible to reduce the number of attention heads in these layers, as fewer heads may be sufficient to learn information. Besides, the decoupling phenomenon also suggests the possibility of more parallelizable architectures in higher layers, which could further improve computational efficiency. Moreover, since the features learned in the lower layers rely mainly on local neighboring tokens, it is also a potential implication to mask far-history tokens to improve model efficiency, if the task does not have strong long-range dependencies.*

## 3 PROBLEM SETUP

**Notation.** For a set $\mathcal{H}$, we use $\Delta(\mathcal{H})$ to denote the set of all probability distributions on $\mathcal{H}$. Let the emission operator $\mathbb{T}^* : \Delta(\mathcal{H}) \to \Delta(\mathcal{O})$. For any $b \in \Delta(\mathcal{H})$, we use $\mathbb{T}^*b \in \Delta(\mathcal{O})$ to denote $\int_{\mathcal{H}} \mathbb{T}^*(x|h)b(h)\mathrm{d}h$. For a vector $a$, we use $[a]_i$ to denote the $i$-th element of $a$. For a sequence $\{x_i\}_{i=1}^{\infty}$, we define the concatenated vector $x_{1:n} = [x_1, \ldots, x_n]^\top$. For a matrix $A \in \mathbb{R}^{d_1 \times d_2}$, we use $[A]_{(i,\cdot)} \in \mathbb{R}^{d_2}$ and $[A]_{(\cdot,j)} \in \mathbb{R}^{d_1}$ to denote the $i$-th row vector and the $j$-th column vector of $A$ respectively, use $[A]_{(i_1:i_2,\cdot)}$ and $[A]_{(\cdot,j_1:j_2)}$ to denote the submatrix consisting of rows $i_1$ through $i_2$, and the submatrix consisting columns $j_1$ through $j_2$ respectively. For a distribution $P : \{e_1, \ldots, e_p\} \to [0, 1]$ supported on the tabular space, we define the vector $P(\cdot) = [P(e_1), \ldots, P(e_p)]^\top$.

We represent each observation as a one-hot vector $o \in \mathbb{R}^{p+1}$. We collect $n$ i.i.d. HMM-generated sequences, each of length $L$. The corresponding hidden states are denoted by $h$, which are unobserved. The token embedding dimension is $D$. We denote by $T$ the number of attention layers in the Transformer after the features have become decoupled.[2]

### 3.1 TRANSFORMER ARCHITECTURE

We begin by describing the framework of Transformers as follows:

**Attention head.** We first recall the definition of the (self-)Attention head $\mathcal{A}ttn(\cdot, Q, K, V)$. With any input matrix $M$,

$$\mathcal{A}ttn\left(M, Q, K, V\right) = \sigma\left(MQK^T M^T\right) MV,$$

where $\{Q, K, V\}$ refer to the Query, Key and Value matrix respectively. The activation function $\sigma(\cdot)$ can be row-wise softmax function[3] or element-wise ReLU function[4].

---

[2]A more detailed notation table is provided in Table 1.

[3]Given a vector input $v$, the $i$-th element of $\mathrm{Softmax}(v)$ is given by $\exp(v_i)/\sum_j \exp(v_j)$.

[4]$\mathrm{ReLU}(x) = \max\{x, 0\}$

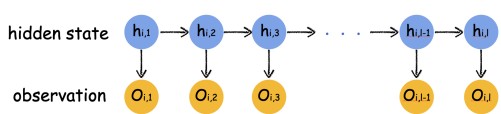

Figure 4: Illustration of Hidden Markov Model.

**Transformer.** Based on the architecture of Attention head, with the input matrix $M$, the definition of multi-head multi-layer Transformer $\text{TF}(\cdot)$ is give by

$$H^{(0)} = M, \quad H^{(l)} = H^{(l-1)} + \sum_{m=1}^{M_l} \mathcal{A}ttn\left(H^{(l-1)}, Q_m, K_m, V_m\right),$$

for any $l \in [N]$, where $N$ refers to the number of Transformer layers, and $M_l$ is the number of Attention heads on the $l$-th layer.

**One-hot encoding.** Considering a vector set with finite elements $\mathcal{S} := \{v_1, v_2, \dots, v_m\}$, the One-hot encoding refers to mapping these vectors into $\mathrm{R}^m$, i.e. $\mathcal{V}ec(\cdot) : \mathcal{S} \to \mathrm{R}^m$. Each vector is mapped to an one-hot vector within $\{e_1, e_2, \dots, e_m\}$, and for any two different vectors $v_s, v_{s'} \in \mathcal{S}$, there will be $\mathcal{V}ec(v_s) \neq \mathcal{V}ec(v_{s'})$.

### 3.2 IN-CONTEXT LEARNING FOR HIDDEN MARKOV MODEL

To show the expressive power of Transformers on sequence tasks, we consider a finite state case in this work, hidden Markov models (HMMs). To perform in-context learning, we collect $n$ i.i.d. demonstrate short observation sequences, i.e. $\{o_{i,1}, \dots, o_{i,L}\}_{i=1}^n$, each sequence consists of $L-1$ observations. Denote the hidden state for each observation as $h_{i,s}$ for any $i \in [n], s \in [L]$, the HMM is defined as (more intuitive description is shown in Figure 4):

$$\mathrm{P}(o_{i,s}|o_{i,1}, \dots, o_{i,s-1}, h_{i,1}, \dots, h_{i,s-1}, h_{i,s}) = \mathrm{P}(o_{i,s}|h_{i,s}), \quad \forall i \in [n], s \in [L],$$
$$\mathrm{P}(h_{i,s}|o_{i,1}, \dots, o_{i,s-1}, h_{i,1}, \dots, h_{i,s-1}) = \mathrm{P}(h_{i,s}|h_{i,s-1}), \quad \forall i \in [n], s \in [L].$$

During testing, to predict $o_{\text{test},k}$ given a long sequence history $\{o_{\text{test},s}\}_{s=1}^{k-1}$, where $k > L$, we construct the input matrix $M_0$ for Transformers in the following format:

$$M_0 := \begin{bmatrix} M_{0,1} & M_{0,2} & \cdots & M_{0,n} & M_{0,\text{test}} \end{bmatrix}^T \in \mathrm{R}^{(n(L+1)+k) \times D},$$

in which the column number $D$ will be specified later, and

$$M_{0,i} := \begin{bmatrix} o_{i,1} & o_{i,2} & \cdots & o_{i,L} & o_{\text{delim}} \\ s_{(i-1)(L+1)+1} & s_{(i-1)(L+1)+2} & \cdots & s_{i(L+1)-1} & s_{i(L+1)} \\ v_{(i-1)(L+1)+1} & v_{(i-1)(L+1)+2} & \cdots & v_{i(L+1)-1} & v_{i(L+1)} \end{bmatrix} \in \mathrm{R}^{D \times (L+1)}, \quad \forall i \in [n],$$

$$M_{0,\text{test}} := \begin{bmatrix} o_{\text{test},1} & o_{\text{test},2} & \cdots & o_{\text{test},k-1} & 0 \\ s_{n(L+1)+1} & s_{n(L+1)+2} & \cdots & s_{n(L+1)+k-1} & s_{n(L+1)+k} \\ v_{n(L+1)+1} & v_{n(L+1)+2} & \cdots & v_{n(L+1)+k-1} & v_{n(L+1)+k} \end{bmatrix} \in \mathrm{R}^{D \times k},$$

where each column of $M_0$, i.e. $[o^T, s^T, v^T]$ represents the embedding for one observation, and $o_{\text{delim}}$ is the delimiter embedding, which represents the end of one sequence. $o \in \mathrm{R}^{p+1}$ refers to the token embedding, which is a one-hot vector within $\{e_1, \dots, e_{p+1}\}$. Specifically, we have

$$o \in \{e_1, \dots, e_p\} \quad \text{for } o \neq o_{\text{delim}}, \quad o_{\text{delim}} = e_{p+1}.$$

The following $s \in \mathrm{R}^2$ is position embedding, which is referred to as

$$[s_{pos}]_1 = \sin\left(\frac{pos}{1000nk}\right), \quad [s_{pos}]_2 = \cos\left(\frac{pos}{1000nk}\right), \quad \forall 1 \leq pos \leq n(L+1)+k.$$

And the last $(D-p-3)$-dim vector $v \in \mathrm{R}^{D-p-3}$ is the fixed embedding, with elements of ones, zeros and indicators for being the test sequence:

$$v_{pos} := \begin{bmatrix} 0_{D-p-5}^\top, 1, 1(pos > n(L+1))^\top \end{bmatrix}^\top, \quad \forall 1 \leq pos \leq n(L+1)+k.$$

We will choose $D \geq 2p^2L$ to allocate sufficient capacity for storing the learned features. After feeding $M_0$ into the Transformer, we will obtain the output $\text{TF}(M_0) \in \text{R}^{(n(L+1)+k)\times D}$ with the same shape as the input, and *read out* the conditional probability $\mathbb{P}(o_{\text{test},k}|o_{\text{test},1:k-1})$ from $[\text{TF}(M_0)]_{(n(L+1)+k,1:p)}$ :

$$\hat{\mathbb{P}}(o_{\text{test},k}|o_{\text{test},1:k-1}) = \text{read}(\text{TF}(M_0)) := [\text{TF}(M_0)]_{(n(L+1)+k,1:p)}.$$

The goal is to predict the conditional probability that is close to the true model.

## 4 THEORETICAL ANALYSIS

### 4.1 LOW-RANK HMM

Our analysis is mainly based on the low-rank structure for HMM.

**Assumption 1** (Low rank structure). *We suppose that the hidden state transition $\mathbb{P} : \mathcal{H} \to \Delta(\mathcal{H})$ admits a low-rank structure: there exist two mappings $w^*, \psi^* : \mathcal{H} \to \mathbb{R}^d$ such that $\mathbb{P}(h'|h) = w^*(h')^\top \psi^*(h)$.*

This condition requires that the latent transition has a low-rank structure, and the underlying representation maps $w^*, \psi^*$ are unknown. This structure is commonly used in representation learning (Agarwal et al., 2020; Uehara et al., 2021; 2022; Guo et al., 2023a).In practice, for example in tabular cases, the transition matrix $P = W\Psi^T$. This condition mean the transition matrix is decomposed into two low-rank matrices, and this low-rank assumption holds in lots of scenarios with high-dimensional data, such as Robot Navigation. The environment information is high-dimensional but the state transition is determined by low-dimension latent common factors.

**Assumption 2** (Over-complete $\gamma$-Observability). *There exists $\gamma > 0$ such that for any distributions $d, d' \in \Delta(\mathcal{H})$, we have $\|\mathbb{T}^*d - \mathbb{T}^*d'\|_1 \geq \gamma\|d - d'\|_1$.*

This condition requires that the observation space is large enough to distinguish the hidden states by observations, i.e., the condition makes the reverse mapping from observation to hidden states a contraction. This aligns with some practical scenarios where meaningful representations allow models to infer latent structure. Observability is necessary and commonly assumed in HMM and partially observed systems (Uehara et al., 2022; Guo et al., 2023a), and it is essentially equivalent to assuming that the emission matrix has full-column rank (Hsu et al., 2012). Further, Assumption 2 implies that we can reverse the inequality to obtain the contraction from observation to hidden state distributions $\|d - d'\|_1 \leq \gamma^{-1}\|\mathbb{T}^*d - \mathbb{T}^*d'\|_1$.

Therefore, we can approximate the posterior hidden state distribution by a posterior sharing the same $(L-1)$-memory (refer to Lemma 4). Together with the low-rank condition that renders the transition $\mathbb{P}(o_k|o_{1:k-1}) := \mu^\top(o_k)\xi(o_{1:k-1})$, where $\mu, \xi$ are two $d$-dim vector functions, we can approximate $\mathbb{P}$ by a $(L-1)$-memory transition in the following lemma: $\hat{\mathbb{P}}_L(o_k|o_{k-1:k-L-1}) := \mu(o_k)^\top \phi(o_{k-L+1:k-1})$, where $\mu(\cdot), \phi(\cdot) \in \mathbb{R}^d$ denote the representations. The representation $\phi$ is a low-rank embedding of the belief distribution of hidden states. For simplicity, here we assume $\phi$ can be represented by a linear mapping.

**Lemma 1** (Model Approximation). *Under Assumptions 1 and 2, there exists a $(L-1)$-memory transition probability $\hat{\mathbb{P}}_L$ with $L = \Theta(\gamma^{-4}\log(d/\epsilon))$ such that*

$$\mathbb{E}_{o_{1:k-1}}\big\|\mathbb{P}(\cdot \mid o_{1:k-1}) - \hat{\mathbb{P}}_L(\cdot \mid o_{k-L+1:k-1})\big\|_1 \leq \epsilon.$$

This lemma shows that for a finite observability coefficient $\gamma$, the model approximation error can be controlled when the memory length $L - 1$ is large enough. To prove this result, we bring the analysis techniques from POMDP literature Guo et al. (2023b); Uehara et al. (2022). The detailed proof can be referred to Appendix F.

### 4.2 MAIN RESULTS

**Assumption 3.** *Given the data observation history, we denote*

$$vo_i = \text{Vectorize}(o_{i,1:L-1}) \in \text{R}^{p(L-1)}, \quad i \in [n],$$

*we define $Z := [vo_1, \ldots, vo_n] \in \mathbb{R}^{p(L-1) \times n}$, then suppose that the mean sample covariance $n^{-1}ZZ^\top$ has lower-bounded eigenvalue: $\lambda_{\min}(n^{-1}ZZ^\top) \geq \alpha$.*

This assumption requires that the eigenvalues of the mean sample covariance are lower-bounded, implying that the data are distributed relatively evenly. It concerns having a sufficiently large number of sequences $n$. This is consistent with practice: modern sequence models are typically trained on large datasets. This condition is commonly used in concentration analysis to bound the generalization error. Our main result can be formally stated as:

**Theorem 1.** *Assume Assumption 1, 2 and 3 hold, there exists a $\mathcal{O}(\ln L + T)$-layer Transformer $\mathrm{TF}_\theta$, such that for any input matrix $M_0$, with probability at least $1 - n^{-1}$ over $\{o_{i,1}, \ldots, o_{i,L}\}_{i=1}^n$:*

$$\mathbb{E}_{o_{\text{test},1:k-1}} \| \mathbb{P}(\cdot | o_{\text{test},1:k-1}) - \mathrm{read}\left(\mathrm{TF}_\theta(M_0)\right) \|_1$$

$$\leq \underbrace{\mathcal{O}(de^{-\gamma^4 L})}_{\text{model approximation}} + \underbrace{\mathcal{O}(pL^{1/2}e^{-\alpha T/(2L)})}_{\text{optimization}} + \underbrace{\mathcal{O}(pL\sqrt{\ln(nLp)}/(\sqrt{n}\alpha) + Ld/\alpha \cdot e^{-L\gamma^4})}_{\text{generalization}}.$$

*More specifically, the Transformer contains $\mathcal{O}(\ln L)$ lower layers and $\mathcal{O}(T)$ upper layers, and the learned features become decoupled after the first $\mathcal{O}(\ln L)$ layers.*

The proof is in Appendix D. Theorem 1 demonstrates that a sufficiently large Transformer can accurately approximate the HMM, revealing its strong expressive power in modeling sequential data.

**Sources of errors.** As shown in Lemma 1, a fixed-length memory model is sufficient to approximate the full-memory transition probabilities, introducing only a small "model approximation" error. Our Transformer construction is based primarily on this approximation, denoted as $\mathbb{P}_L$. The "generalization" error arises due to the use of a finite sample size $n$: we learn $\mathbb{P}_L$ from $n$ i.i.d. samples, and the optimal learned model we can obtain, $\hat{\mathbb{P}}_L$, remains close to $\mathbb{P}_L$ as long as $n$ is sufficiently large. The final source of error, the "optimization" error, stems from the finite capacity of the Transformer. Since we approximate $\hat{\mathbb{P}}_L$ using a Transformer with a limited number of layers, a gap between the two remains. However, this gap can be made arbitrarily small by increasing the model size (e.g., number of layers), thereby improving the approximation accuracy.

**Remark 3** (The connection between theory and empirical results). *Consider the layerwose modeling, our explicit construction aligns closely with the empirical observations presented in Section 2. The construction proceeds in several stages. First, in the lower layers, the Transformer learns information from the neighborhood $L$ tokens, gradually incorporating information from nearby to more distant tokens, which is consistent with the patterns shown in Figure 3b. In the upper layers, to take the final prediction, the learned features become decoupled and are used to infer a causal structure aligned with the underlying HMM task, which corresponds to Figure 2 and the rising-then-falling trend observed in Figure 3b. Finally, the overall progression—from token-level feature learning to task-level abstraction—matches the trends in Figure 3a, reflecting a clear layerwise hierarchy in how Transformers process sequential information.*

**Discussion on induction head.** The "induction head" phenomenon demonstrates that Transformers can learn to predict future tokens by identifying repeating patterns in the input sequence. In contrast, our result reveals that even when such patterns do not appear in the input history, the Transformer can still make accurate predictions by learning to infer, rather than simply matching previous patterns. This highlights a deeper aspect of its in-context learning ability. As a result, our approach remains effective even with a relatively small sample size. Moreover, when the sample size is sufficiently large, our framework becomes consistent with the induction head behavior, bridging the two perspectives.

## 4.3 EXTENSION TO INDISTINGUISHABLE SITUATION

In NLP tasks, a natural assumption is that the cardinality of hidden state space may be larger that the observation space evidence, or the true number of observations that can reveal the hidden states is small, called "weak revealing" cases. In this section, we show that Transformer can still perform well under such ambiguous setting. Inspired by the overcomplete POMDPs (Liu et al., 2022b), we start by expanding the output space of emission operators.

**Assumption 4** (Under-complete $\gamma$-Observability). *Let operator $\mathbb{M} : \Delta(\mathcal{H}) \to \Delta_m(\mathcal{O} \times \cdots \times \mathcal{O})$ such that $\mathbb{M}d_{\mathcal{H}} : \mathcal{O} \times \cdots \times \mathcal{O} \to \mathbb{R}$ denotes $\int_{\mathcal{O} \times \cdots \times \mathcal{O}} \mathbb{M}(o_{t:t+m}|h_t)d_{\mathcal{H}}(h_t)\mathrm{d}h_t$, where $m$ is a small*

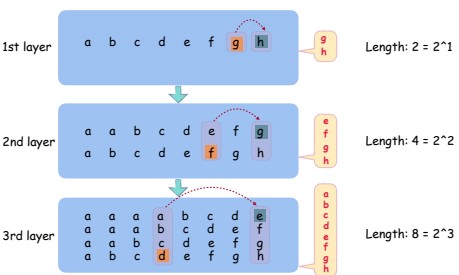

Figure 5: Illustration of Feature learning process.

constant such that $m < L$. There exists $\tilde{\gamma} > 0$ such that for any distributions $d, d' \in \Delta(\mathcal{H})$, we have $\|\mathbb{M}b - \mathbb{M}b'\|_1 \geq \tilde{\gamma}\|b - b'\|_1$.

Then the corresponding theorem should be:[5]

**Theorem 2.** *Denote the data observation* $Z' := [o_{1,1:L-m}, \ldots, o_{n,1:L-m}] \in \mathbb{R}^{p(L-m) \times n}$. *Assume Assumption 1, 4 hold, and* $\lambda_{\min}(n^{-1}Z'Z'^T) \geq \alpha$, *there exists a* $\mathcal{O}(\ln L + T)$-*layer Transformer* $\mathrm{TF}_\theta$, *such that for any input matrix* $M_0$, *with probability at least* $1 - n^{-1}$ *over* $\{o_{i,1}, \ldots, o_{i,L}\}_{i=1}^n$:

$$
\mathbb{E}_{o_{\text{test},1:k-1}} \|\mathbb{P}(\cdot|o_{\text{test},1:k-1}) - \mathrm{read}(\mathrm{TF}_\theta(M_0))\|_1
$$
$$
\leq \underbrace{\mathcal{O}(de^{-\tilde{\gamma}^4 L})}_{model\ approximation} + \underbrace{\mathcal{O}(p^m L^{1/2} e^{-\alpha T/(2L)})}_{optimization} + \underbrace{\mathcal{O}(p^m L\sqrt{\ln(nLp)}/(\sqrt{n}\alpha) + Ld/\alpha \cdot e^{-L\tilde{\gamma}^4})}_{generalization}.
$$

The proof is in Appendix E. From Theorem 2, we show that Transformers can still learn HMMs efficiently under such "weak revealing" case, by concatenating several steps of future observations.

## 5 TRANSFORMER CONSTRUCTION AND PROOF SKETCHES

### 5.1 PROOF SKETCHES FOR THEOREM 1

Recalling Lemma 1, our Transformer construction is mainly based on approximating $\mathbb{P}_L(\cdot|o_{\text{test},k-L+1:k-1})$ with expression: $\mathbb{P}_L(o_k|o_{k-L+1:k-1}) = \mu^\top(o_k)\phi(o_{k-L+1:k-1})$.

To approximate the error in prediction, we can take the following decomposition:

$$
\mathbb{E}_{o_{\text{test},1:k-1}} \|\mathbb{P}(\cdot|o_{\text{test},1:k-1}) - \mathrm{read}(\mathrm{TF}_\theta(M_0))\|_1
$$
$$
\leq \underbrace{\mathbb{E}_{o_{\text{test},1:k-1}} \|\mathbb{P}(\cdot|o_{\text{test},1:k-1}) - \mathbb{P}_L(\cdot|o_{\text{test},k-L+1:k-1})\|_1}_{\epsilon_1:\ model\ approximation}
$$
$$
+ \underbrace{\mathbb{E}_{o_{\text{test},1:k-1}} \|\mathbb{P}_L(\cdot|o_{\text{test},k-L+1:k-1}) - \hat{\mathbb{P}}_L(\cdot|o_{\text{test},k-L+1:k-1})\|_1}_{\epsilon_2:\ generalization} \quad (1)
$$
$$
+ \underbrace{\mathbb{E}_{o_{\text{test},1:k-1}} \|\hat{\mathbb{P}}_L(\cdot|o_{\text{test},k-L+1:k-1}) - \mathrm{read}(\mathrm{TF}_\theta(M_0))\|_1}_{\epsilon_3:\ optimization},
$$

where $\hat{\mathbb{P}}_L(\cdot|o_{\text{test},k-L+1:k-1}) \in \mathbb{R}^p$ refers to the optimal approximation for $\mathbb{P}_L$ based on $n$ i.i.d. samples we collected. Considering the one-hot format of $o_k$ and the linear assumption on $\phi(\cdot)$, we can express both $\mu(\cdot)$ and $\phi(\cdot)$ as linear function, which implies that

$$
\mathbb{P}_L(\cdot|o_{k-L+1:k-1}) := W_* o_{k-L+1:k-1},
$$

---

[5]The conditional probability in Theorem 2 is related to a $m$-step prediction, which induces that the cardinality of observation is $p^m$. So we enlarge $D$ such that $D \geq 2p^m L$, and the read out function should be $\hat{\mathbb{P}}(o_{\text{test},k}|o_{\text{test},1:k-1}) = \mathrm{read}(\mathrm{TF}(M_0)) := [\mathrm{TF}(M_0)]_{(n(L+1)+k,(L+1)(p+3)+1:(L+1)(p+3)+p^m)}$.

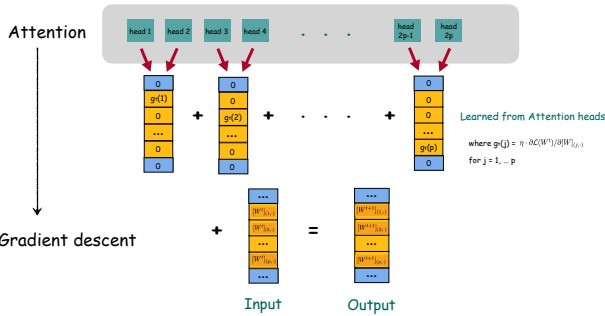

Figure 6: Illustration of gradient descent performance.

for some $W_* \in \mathbb{R}^{p \times p(L-1)}$[6]. Accordingly, we have $\hat{\mathbb{P}}_L(\cdot|o_{\text{test},k-L+1:k-1}) := \hat{W} o_{\text{test},k-L+1:k-1}$, in which

$$\hat{W} := \arg\min_W \mathcal{L}(W) := \arg\min_W \sum_i \|o_{i,L} - W z_i\|_2^2. \tag{2}$$

Here we use the short-hand notation $z_i := o_{i,1:L-1} \in \mathbb{R}^{p(L-1)}$. From Lemma 1, we obtain $\epsilon_1 = \mathcal{O}(de^{-\gamma^4 L})$. And in the following analysis, we focus on bounding $\epsilon_2$ and $\epsilon_3$, respectively.

### 5.1.1 TRANSFORMER CONSTRUCTION

To predict the conditional probability vector $\hat{\mathbb{P}}_L(\cdot|o_{\text{test},k-L+1:k-1})$, the transformer proceeds in three main steps: (i) it first learns the $(L-1)$-step history feature $o_{i,1:L-1}$ associated with $o_{i,L}$, as well as $o_{\text{test},k-L+1:k-1}$ associated with $o_{\text{test},k}$, (ii)it then performs linear regression based on Eq. (6), (iii)finally, it approximates $\hat{\mathbb{P}}_L(\cdot|o_{\text{test},k-L+1:k-1})$ using $\hat{W}$ and $o_{\text{test},k-L+1:k-1}$. The explicit construction of the Transformer is detailed below:

**Decoupled feature learning.** Before formally construction, for any step index $1 \le r < L$, we define history and future matrix $Z_r, F_r \in \mathbb{R}^{(n(L+1)+k) \times (p+3)}$ for further analysis:

$$[Z_r]_{(t,\cdot)} := \begin{cases} [M_0]_{(t-r,1:p+3)}, & r < t \le n(L+1)+k, \\ [M_0]_{(1,1:p+3)}, & 1 \le t \le r, \end{cases}$$

$$[F_r]_{(t,\cdot)} := \begin{cases} [M_0]_{(t+r,1:p+3)}, & 1 \le t \le n(L+1)+k-r, \\ [M_0]_{(n(L+1)+k,1:p+3)}, & n(L+1)+k-r < t \le n(L+1)+k. \end{cases}$$

To be specific, for each $o_{i,s}$, $Z_r$ and $F_r$ are corresponding to $o_{i,s-r}$ (history observation) and $o_{i,s+r}$ (future observation) respectively. To learn these two types of features, we use two special matrices on the position embedding vector of each observation:

$$A := \beta_1 \begin{bmatrix} \cos(\frac{1}{1000nk}) & \sin(\frac{1}{1000nk}) \\ -\sin(\frac{1}{1000nk}) & \cos(\frac{1}{1000nk}) \end{bmatrix}, \quad B := \beta_1 \begin{bmatrix} \cos(\frac{1}{1000nk}) & -\sin(\frac{1}{1000nk}) \\ \sin(\frac{1}{1000nk}) & \cos(\frac{1}{1000nk}) \end{bmatrix}.$$

For $t_1, t_2 \in [1 : n(L+1)+k]$ with position embedding vectors $s_{t_1}, s_{t_2}$, we have

$$s_{t_1}^T A s_{t_2} = \beta_1 \cdot \cos\left(\frac{t_1 - t_2 - 1}{1000nk}\right), \quad s_{t_1}^T B s_{t_2} = \beta_1 \cdot \cos\left(\frac{t_1 - t_2 + 1}{1000nk}\right).$$

By using A in Query-Key matrix with enough large $\beta_1$, and applying the softmax activation along with a carefully designed Value matrix, we can learn $Z_1$ after the first Attention layer. On the second layer, we again use $A$ to design Query-Key matrix, which enables the learning of $Z_2, Z_3$ (see Figure 5 as a detailed illustration). Repeating such process for $\mathcal{O}(\ln L)$ layers, we will obtain $\{Z_1, \ldots, Z_{L-1}\}$ using $\mathcal{O}(\ln L)$-layer single-head Attention. Also, use matrix $B$, we can obtain $F_1$ on the following layer. The output matrix after these decoupled-feature layers should be

$$M_{\text{dec}} = [[M_0]_{(\cdot,:p+3)}, Z_1, Z_2, Z_3, \ldots, Z_{L-1}, F_1, [M_0]_{(\cdot,(L+1)(p+3)+1:D)}] \in \mathbb{R}^{(n(L+1)+k) \times D}.$$

---

[6]As the $(p+1)$-th dimension is designed only for $o_{\text{delim}}$, we consider the observation as a $p$-dim vector for simplicity.

**Gradient descent performing and final prediction.** The following $\mathcal{O}(T)$-layer architecture is designed to learn $\hat{\mathbb{P}}_L(\cdot|z_k)$ based on history information $\{Z_1, \ldots, Z_{L-1}\}$. To be specific, from Eq. (2), we need to take linear regression to estimate a matrix $\hat{W} \in \mathbb{R}^{p \times p(L-1)}$. To perform such estimation process for $W$, we construct a $2p$-head $\mathcal{O}(T)$-layer Attention. Each layer can perform single gradient descent step on $\mathcal{L}(W)$, starting from an initial value 0. Each row of $W$ is assigned to two independent attention heads for parallel learning (see Figure 6 for detailed illustration). The construction closely follows the method proposed in Bai et al. (2024), with the key difference being that we use $F_1$ to pick up $n$ samples for the gradient descent updating. After $\mathcal{O}(T)$-step gradient descent, we use the learned $\{[\hat{W}]_{(1,\cdot)}, \ldots, [\hat{W}]_{(p,\cdot)}\}$ and $o_{\text{test},k-L+1:k-1}$ to predict $\hat{\mathbb{P}}_L(\cdot|o_{\text{test},k-L+1:k-1})$. The corresponding error $\epsilon_3 = \mathcal{O}(pL^{1/2}e^{-\alpha T/(2L)})$ can be estimated using Lemma 7.

### 5.1.2 Generalization Error Approximation

Using the notations for labels and covariates $O := [o_{1,L}, \ldots, o_{n,L}] \in \mathbb{R}^{p \times n}$, $Z = [o_{1,1:L-1}, \ldots, o_{n,1:L-1}] \in \mathbb{R}^{p(L-1) \times n}$, the least square estimator has the following closed-form solution: $\hat{W} := OZ^T(ZZ^T)^{-1}$.

Then, denoting $z_{\text{test}} := o_{\text{test},k-L+1:k-1}$ and error $\Delta := O - W_* Z$, we can take the estimator into $\epsilon_2$ and upper bound it by

$$
\begin{aligned}
\epsilon_2 &\leq \sum_{j=1}^p \sqrt{L} \|[W_*]_{(j,\cdot)} - [O]_{(j,\cdot)} Z^T (ZZ^T)^{-1}\|_2 \leq \frac{\sqrt{L}}{n\alpha} \sum_{j=1}^p \|[\Delta]_{(j,\cdot)} Z^T\|_2 \\
&\leq \frac{\sqrt{L}}{n\alpha} \sum_{j=1}^p \|([\Delta]_{(j,\cdot)} - \mathbb{E}[[\Delta]_{(j,\cdot)}]) Z^T\|_2 + \frac{\sqrt{L}}{n\alpha} \sum_{j=1}^p \|\mathbb{E}[[\Delta]_{(j,\cdot)}]) Z^T\|_2
\end{aligned}
\tag{3}
$$

where the second inequality uses the definition $O = \Delta + W_* Z$ and $\lambda_{\min}(ZZ^\top) \geq \alpha$ in Assumption 3, and invokes the Cauchy-Schwartz inequality. For the first term on the last row of (3), we use the matrix concentration in Lemma 6 to obtain that with a high probability,

$$
\|([\Delta]_{(j,\cdot)} - \mathbb{E}[[\Delta]_{(j,\cdot)}]) Z^T\|_2 \leq \mathcal{O}\big(\sqrt{nL\ln(nLp^2)}\big).
$$

For the second term on the last row of (3), based on the observation that $\mathbb{E}[[\Delta]_{(j,i)}] = \mathbb{E}_{o_{i,1:k-1}}[\mathbb{P}(e_j \mid o_{1:k-1}) - \mathbb{P}_L(e_j \mid o_{k-L+1:k-1})]$, we can bound it by $\mathcal{O}(Ld/\alpha \cdot e^{-L\gamma^4})$ via Lemma 1.

### 5.2 Proof Sketches for Theorem 2

The error analysis and the corresponding Transformer construction follow a similar approach to Theorem 2, with one key modification. After the decoupled feature extraction stage, the resulting output matrix takes the following form:

$$
M_{\text{dec}} = [[M_0]_{(\cdot,1:p+3)}, Z_1, Z_2, \ldots, Z_{L-m}, F_1, F_2, \ldots, F_m, [M_0]_{(\cdot,(L+1)(p+3)+1:D)}].
$$

Before feeding it into subsequent Attention layers, we apply an one-hot encoding function $\mathcal{V}ec(\cdot)$ to each row of $\{[M_0]_{(\cdot,1:p)}, [F_1]_{(\cdot,1:p)}, \ldots, [F_{m-1}]_{(\cdot,1:p)}\}$, which correspond to the current and future observations at each time step.

## 6 Conclusion

This work advances our theoretical and empirical understanding of how Transformers achieve strong generalization across diverse sequence learning tasks. By analyzing their layerwise behavior and constructing explicit architectures for modeling HMMs, we demonstrate that Transformers gradually transition from learning local, token-level features in lower layers to forming decoupled representations in upper layers. These findings align with empirical observations, as well as providing a principled explanation for the Transformer's expressiveness and efficiency in multi-task and in-context learning settings.

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

# A    RELATED WORKS

**Expressiveness of Transformer.**    The expressive power of Transformers has been studied extensively from various perspectives. For example, Akyürek et al. (2022); Von Oswald et al. (2023); Mahankali et al. (2023); Dai et al. (2022) demonstrate that a single attention layer is sufficient to compute a single gradient descent step. Garg et al. (2022); Bai et al. (2024); Guo et al. (2023b) show that Transformers can implement a wide range of machine learning algorithms in context. Similarly, Xie et al. (2021); Wang et al. (2023); Jiang (2023) establish that Transformers can approximate Bayesian optimal inference. Other works have explored different capabilities of Transformers: Liu et al. (2022a) show they can learn shortcuts to automata, Lin et al. (2023) demonstrate their ability to implement reinforcement learning algorithms, and Nichani et al. (2024) reveal their capacity to learn Markov causal structures under a fixed transition matrix, Sander et al. (2024); Wu et al. (2025) show the expressiveness power on learning autoregressive models.

**Hidden Markov Model.**    Identification for uncontrolled partially observable systems has been broadly studied, especially for the spectral learning based models (Hsu et al., 2012; Van Overschee & De Moor, 1995; Song et al., 2010; Hamilton et al., 2013; Kulesza et al., 2015). Intuitively, all the frameworks require some observability conditions to reveal the hidden states via sufficient observations. For complex sequential spaces with a large hidden state space, there is another line of work considering structured latent transitions, allowing for more efficient inference and computation complexity (Siddiqi & Moore, 2005; Felzenszwalb et al., 2003; Dedieu et al., 2019; Siddiqi et al., 2010; Chiu et al., 2021). Especially, Chiu et al. (2021) consider a low-rank structure for hidden state transitions. Such a low-rank structure is also widely studied in partially observable Markov Decision processes (Uehara et al., 2022; Guo et al., 2023a; Zhong et al., 2022; Wang et al., 2022; Zhan et al., 2022). The most related ones to our work are Uehara et al. (2022); Guo et al. (2023a), which utilize the low-rank latent transition and observability to avoid a long-memory learning and inference. Instead, they can approximate the posterior distribution of the hidden states given whole observations by a distribution conditioned on a fixed-size history.

**Transformer and Markov Data.**    A growing body of work studies Transformers through the lens of Markovian structures and in-context learning. Bietti et al. (2023) interpret Transformers as dynamic memory systems that integrate features across layers. Edelman et al. (2024) and Ekbote et al. (2025) analyze induction heads, showing that Transformers can implement Markov chains and pattern-matching behaviors. Zhou et al. (2024) demonstrate that Transformers can learn variable-order Markov chains in-context. Makkuva et al. (2024a;b) provide principled frameworks to analyze attention on Markov data and study how learning dynamics evolve from local to global representations. Rajaraman et al. (2024) show that constant-depth Transformers suffice to model Markov processes. Nichani et al. (2024) study how Transformers learn causal structures, while Li et al. (2023) and Ren & Liu (2024) focus on topic structure and representation learning dynamics in in-context learning. Our work differs in three main aspects. First, rather than studying training dynamics or pattern-matching mechanisms, we focus on the expressive power of Transformers for representing hidden Markov models. Second, we observe a feature-decoupling phenomenon, in which Transformers can infer latent states even without repeated patterns in the input, contrasting with classical induction head behavior that relies on explicit token matches. Third, while our approach works with relatively small sample sizes, it becomes consistent with induction-head behavior when the sample size is large, bridging the inference-driven and pattern-matching perspectives.

# B    ADDITIONAL EXPERIMENT DETAILS AND RESULTS

## B.1    EXPERIMENT SETTINGS

Here we construct a dataset generated by a mixture of Hidden Markov Models (HMMs). Each HMM is used to model a tasks-specific distribution, and by mixing them we get a dataset similar to a pre-training corpus to learn language modeling on. In specific, we randomly simulate 8192 HMMs. The generation process is as follows. There is an initial task distribution on which we sample the HMM id. Each HMM composes of 128 hidden states randomly transiting between each other. Each next state depends purely on the previous state, making the sequence of hidden states Markovian. All HMMs share a 16-token vocabulary. Each hidden state is associated with an emission distribution

to randomly output a token. We sample 131k data, which allows training for 64 epochs, with 64 steps in each epoch on a batch size of 32. We build a transformer of 16 layers and 16 heads in each layer, and a hidden state dimension of 1024. The experiments run on a single V100 GPU with 16 GB of memory for 10 hours. The mixture-of-HMMs simulation runs with default multiprocessing of Python.

See Figure 7 for the attention heatmap.

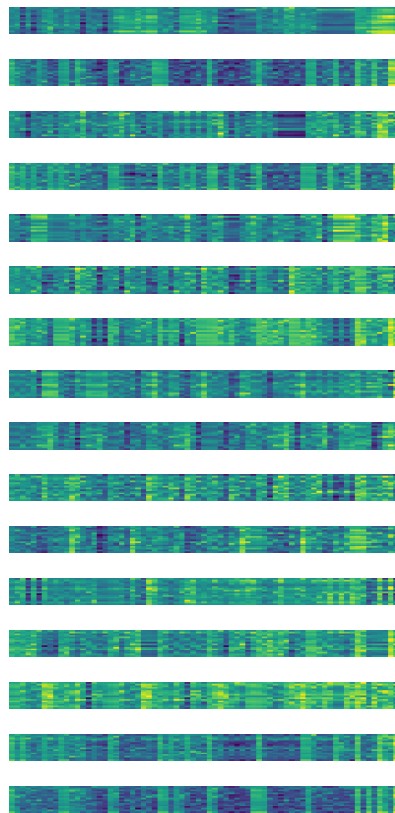

Figure 7: Attention of the Transformer on in-context learning inputs. The y-axis denotes layers and attention heads within each layers, and the x-axis denotes the attention of the last token on all previous tokens in the ICL input (including both demonstrative examples and the test input).

## B.2 Additional results on other models

**Verification on smaller models.** We conducted additional experiments on smaller models. We use the same experimental setting and investigate Transformers of smaller sizes (number of layers 8, number of heads 8) and (number of layers 4, number of heads 4). The 8-layer model is capable of learning the HMMs with the final-example accuracy of 0.707 (a similar level to the 16-layer model, indicating a saturated accuracy). In contrast, the 4-layer model has a degraded accuracy of 0.213, meaning that the learning ability gradually emerges between a layer depth of 4 and 8. Moreover, interestingly, we observed a similar feature decoupling phenomenon. The results of 8-layer 8-head Transformer can be seen in Figure 8, 9, 10 and 11. The results of 4-layer 4-head Transformer can be seen in Figure 12, 13, 14 and 15.

**Verification on larger model.** We analyze the LLaMA-3-8B model on the SST-2 dataset using 64 (demonstration set, test sample) pais, each with 16 samples of length 16. We apply 16 random permutations per group and measure attention consistency across permutations using the metric 1 - std/mean of attention logits to the final token. The results (unfortunately we are prohibited from uploading images) reveal a clear trend: higher layers contain a larger proportion of position-invariant

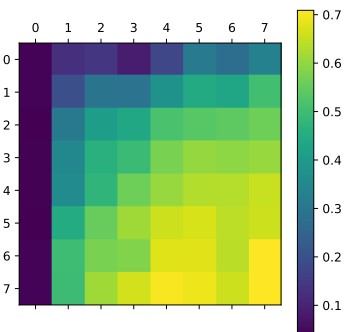

Figure 8: Accuracy of the Transformer under in-context learning setting.

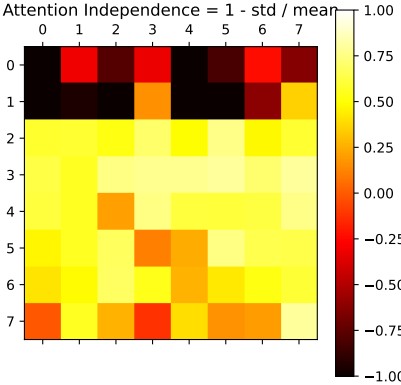

Figure 9: After randomly shuffling the positions of demonstrative inputs, we examine how the logits receive changes over layers (y-axis) and attention heads (x-axis). The measure is $1 - \frac{\text{std(logits)}}{\text{mean(logits)}}$.

heads, suggesting these layers rely less on the absolute positions of ICL examples. More specifically, the initial 8 layers have an average ratio of std / mean = 1.59, while the last 8 layers have the average ratio of 0.79. See reults in Figure 16.

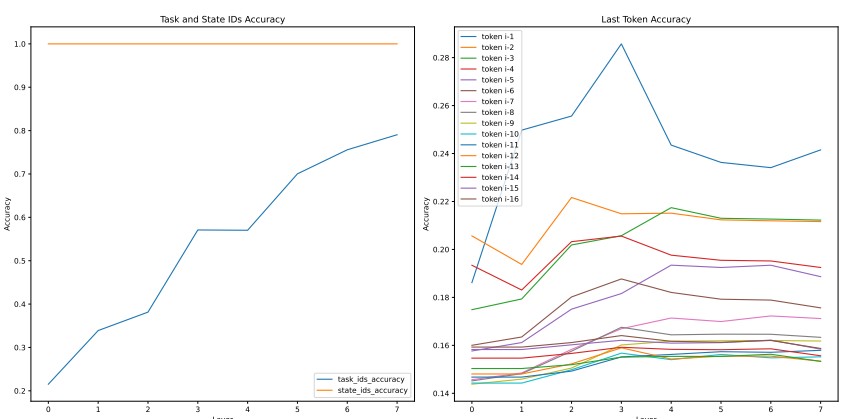

Figure 10: Investigation on Transformer recognitions.

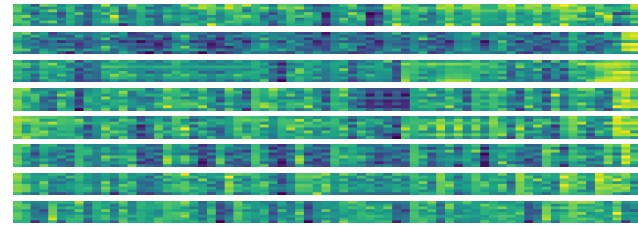

Figure 11: Attention of the Transformer on in-context learning inputs.

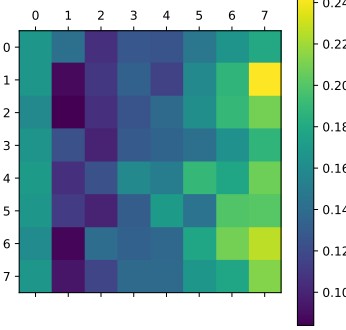

Figure 12: Accuracy of the Transformer under in-context learning setting.

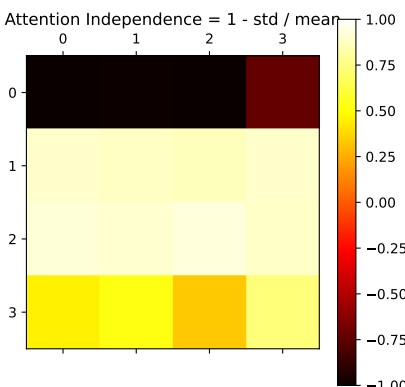

Figure 13: After randomly shuffling the positions of demonstrative inputs, we examine how the logits receive changes over layers (y-axis) and attention heads (x-axis). The measure is $1 - \frac{\text{std(logits)}}{\text{mean(logits)}}$.

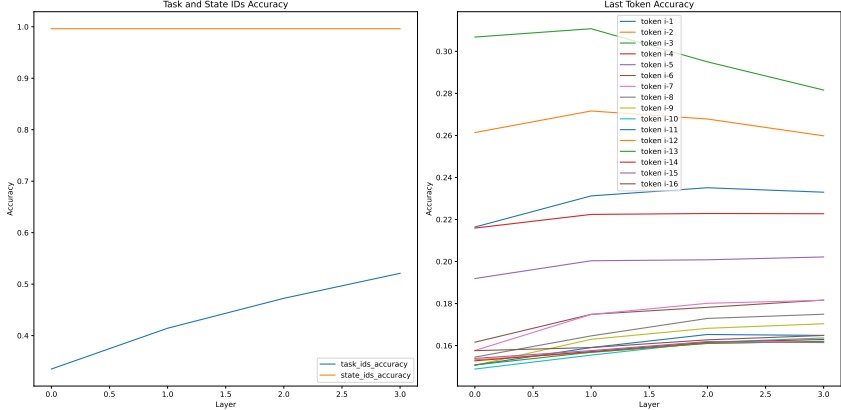

Figure 14: Investigation on Transformer recognitions.

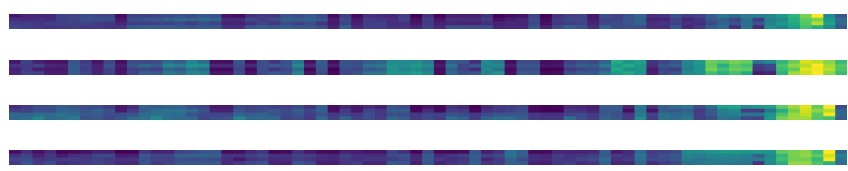

Figure 15: Attention of the Transformer on in-context learning inputs.

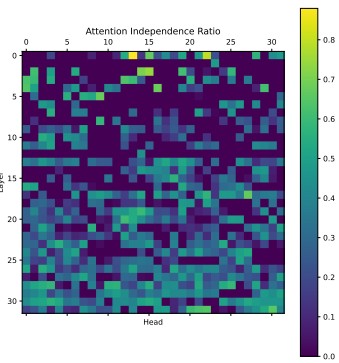

Figure 16: After randomly shuffling the positions of demonstrative inputs, we examine how the logits receive changes over layers (y-axis) and attention heads (x-axis). The measure is $1 - \frac{\text{std(logits)}}{\text{mean(logits)}}$.

## C  NOTATION TABLE

Table 1: The table of notations used in this paper.

| Notation | Description |
|---|---|
| $\Delta(\mathcal{H})$ | the set of all probability distributions on $\mathcal{H}$ |
| $\mathbb{T}^*$ | the emission operator |
| $\mathbb{T}^*b$ | $\int_{\mathcal{H}} \mathbb{T}^*(x\|h)b(h)\mathrm{d}h$ |
| $e_j$ | one-hot vector |
| $[a]_j$ | the $i$-th element of vector $a$ |
| $x_{1:n}$ | concatenated vector $[x_1, \ldots, x_n]^\top$ |
| $[A]_{(i,\cdot)}$ | the $i$-th row vector of $A$ |
| $[A]_{(\cdot,j)}$ | the $j$-th column vector of $A$ |
| $[A]_{(i_1:i_2,\cdot)}$ | the submatrix consisting of rows $i_1$ through $i_2$ of $A$ |
| $[A]_{(\cdot,j_1:j_2)}$ | the submatrix consisting columns $j_1$ through $j_2$ of $A$ |
| $P(\cdot)$ | the vector $[P(e_1), \ldots, P(e_p)]^\top$ for a distribution $P : \{e_1, \ldots, e_p\} \to [0,1]$ |
| $L$ | sequence length on training samples |
| $\gamma$ | observability coefficient |
| $p$ | observation state number |
| $d$ | feature dimension in transition matrix low-rank structure |
| $n$ | sequence sample number |
| $k$ | sequence length on test sample |
| $T$ | the number of gradient descent steps after feature obtaining |

## D  PROOFS FOR THEOREM 1

Recalling Lemma 1, our Transformer construction is mainly based on approximating $\mathbb{P}_L(\cdot|o_{\text{test},k-L+1:k-1})$ with expression:

$$\mathbb{P}_L(o_k|o_{k-L+1:k-1}) = \mu(o_k)^T\phi(o_{k-L+1:k-1}).$$

To approximate the error in prediction, we can take the following decomposition:

$$\mathbb{E}_{o_{\text{test},1:k-1}}\|\mathbb{P}(\cdot|o_{\text{test},1:k-1}) - \text{read}(\text{TF}_\theta(M_0))\|_1$$

$$\leq \underbrace{\mathbb{E}_{o_{\text{test},1:k-1}}\|\mathbb{P}(\cdot|o_{\text{test},1:k-1}) - \mathbb{P}_L(\cdot|o_{\text{test},k-L+1:k-1})\|_1}_{\epsilon_1:\text{model approximation}}$$

$$+ \underbrace{\mathbb{E}_{o_{\text{test},1:k-1}}\|\mathbb{P}_L(\cdot|o_{\text{test},k-L+1:k-1}) - \hat{\mathbb{P}}_L(\cdot|o_{\text{test},k-L+1:k-1})\|_1}_{\epsilon_2:\text{generalization}} \quad (4)$$

$$+ \underbrace{\mathbb{E}_{o_{\text{test},1:k-1}}\|\hat{\mathbb{P}}_L(\cdot|o_{\text{test},k-L+1:k-1}) - \text{read}(\text{TF}_\theta(M_0))\|_1}_{\epsilon_3:\text{optimization}},$$

where $\hat{\mathbb{P}}_L(\cdot|o_{\text{test},k-L+1:k-1}) \in \mathbb{R}^p$ refers to the optimal approximation for $\mathbb{P}_L$ based on $n$ i.i.d. samples we collected.

Considering the one-hot vector $o_k \in \mathbb{R}^p$, which representing the observation state[7], we can express $\mu(\cdot)$ as

$$\mu(o_k) = Uo_k,$$

for some $U \in \mathbb{R}^{d\times p}$. Also, recalling the linear mapping assumption for $\phi(\cdot)$, we can also obtain

$$\phi(o_{k-L+1:k-1}) = Vo_{k-L+1:k-1},$$

for some $V \in \mathbb{R}^{d\times p(L-1)}$, which further implies that

$$\mathbb{P}_L(o_k|o_{k-L+1:k-1}) = o_k^T U^T V o_{k-L+1:k-1}.$$

---

[7]As the $(p+1)$-th dimension is designed only for $o_{\text{delim}}$, we consider the observation as a $p$-dim vector in proofs for simplicity.

As the feature embeddings are within $\{e_1, \ldots, e_p\}$, the vector $\mathbb{P}_L(\cdot|o_{k-L+1:k-1}) \in \mathbb{R}^p$ equals to

$$\mathbb{P}_L(\cdot|o_{k-L+1:k-1}) = U^T V o_{k-L+1:k-1} := W_* o_{k-L+1:k-1}, \tag{5}$$

where $W_* \in \mathbb{R}^{p \times p(L-1)}$. So for $\hat{\mathbb{P}}_L(\cdot|o_{\text{test},k-L+1:k-1}) := \hat{W} o_{\text{test},k-L+1:k-1}$, the solution is

$$\hat{W} := \arg\min_W \mathcal{L}(W) := \arg\min_W \sum_i \|o_{i,L} - W z_i\|_2^2, \tag{6}$$

where we use the short-hand notation $z_i := o_{i,1:L-1} \in \mathrm{R}^{p(L-1)}$. From Lemma 1, we have that $\epsilon_1 = \mathcal{O}(de^{-\gamma^4 L})$. In the following two subsections, we focus on bounding $\epsilon_2$ and $\epsilon_3$, respectively.

## D.1 TRANSFORMER CONSTRUCTION

To approximate the conditional probability vector $\hat{\mathbb{P}}_L(\cdot|o_{\text{test},k-L+1:k-1})$, the transformer mainly takes three steps: (1) firstly learning the $(L-1)$-step history features $o_{i,1:L-1}$ for $o_{i,L}$, as well as $o_{\text{test},k-L+1:k-1}$ for $o_{\text{test},k}$, (2) then performing linear regression based on Eq. (6), (3) finally approximating $\hat{\mathbb{P}}_L(\cdot|o_{\text{test},k-L+1:k-1})$ using $\hat{W}$ and $o_{\text{test},k-L+1:k-1}$. The explicit construction of the Transformer is as follows:

**Decoupled feature learning.** Here we first construct an $\mathcal{O}(\ln L)$-layer single head Attention, to learn $o_{i,1:L-1}$ for $o_{i,L}$, as well as $o_{\text{test},k-L+1:k-1}$ for $o_{\text{test},k}$. Before formally construction, for any step index $1 \le r < L$, we define history and future matrix $Z_r, F_r \in \mathbb{R}^{(n(L+1)+k) \times (p+3)}$ for further analysis:

$$[Z_r]_{(t,\cdot)} := \begin{cases} [M_0]_{(t-r,1:p+3)}, & r < t \le n(L+1)+k, \\ [M_0]_{(1,1:p+3)}, & 1 \le t \le r, \end{cases}$$

$$[F_r]_{(t,\cdot)} := \begin{cases} [M_0]_{(t+r,1:p+3)}, & 1 \le t \le n(L+1)+k-r, \\ [M_0]_{(n(L+1)+k,1:p+3)}, & n(L+1)+k-r < t \le n(L+1)+k, \end{cases}$$

Here we also define a special matrix

$$A := \beta_1 \begin{bmatrix} \cos(\frac{1}{1000nk}) & \sin(\frac{1}{1000nk}) \\ -\sin(\frac{1}{1000nk}) & \cos(\frac{1}{1000nk}) \end{bmatrix},$$

where $\beta_1 > 0$ is a fixed constant. Then on the first layer, the Query-Key matrix is designed as

$$QK^{(1)} := \begin{bmatrix} 0_{(p+1)\times(p+1)} & 0 & 0 \\ 0 & A & 0 \\ 0 & 0 & 0 \end{bmatrix} \in \mathbb{R}^{D \times D},$$

which induces that with input matrix $M_0$, we have

$$[M_0]_{(t_1,\cdot)}^T QK^{(1)} [M_0]_{(t_2,\cdot)} = \beta_1 \cdot \cos\left(\frac{t_1 - t_2 - 1}{1000nk}\right),$$

for any $1 \le t_1, t_2 \le n(L+1)+k$. Then with softmax function on $M_0 QK^{(1)} M_0^T$, as well as the Value matrix

$$V^{(1)} := \begin{bmatrix} 0_{(p+3)\times(p+3)} & I_{(p+3)\times(p+3)} & 0_{(p+3)\times(D-2p-6)} \\ 0 & 0 & 0 \\ 0 & 0 & 0 \end{bmatrix} \in \mathbb{R}^{D \times D},$$

sending $\beta_1 \to \infty$, we obtain the output on each row as

$$\left[\text{Softmax}\left([M_0]_{(t,\cdot)} QK^{(1)} M_0^T\right) M_0 V^{(1)}\right]_{(t,\cdot)} = [0, [M_0]_{(t-1,1:p+3)}, 0]^T, \quad \forall 1 < t \le n(L+1)+k,$$

which refers that after the first Attention layer, the output matrix should be

$$M_1 = M_0 + \mathcal{A}ttn(M_0, QK^{(1)}, V^{(1)}) = [[M_0]_{(\cdot,1:p+3)}, Z_1, [M_0]_{(\cdot,2(p+3)+1:D)}].$$

It implies that the first layer Attention head learn the first history feature $o_{i,L-1}$ for each observation $o_{i,L}$. Then on the second layer, we design the Query-Key matrix as

$$QK^{(2)} := \begin{bmatrix} 0_{(2p+4)\times(p+1)} & 0_{(2p+4)\times 2} & 0_{(2p+4)\times(D-p-3)} \\ 0_{2\times(p+1)} & A & 0_{2\times(D-p-3)} \\ 0_{(D-2p-6)\times(p+1)} & 0_{(D-2p-6)\times 2} & 0_{(D-2p-6)\times(D-p-3)} \end{bmatrix},$$

as well as the Value matrix as

$$V^{(2)} := \begin{bmatrix} 0_{2(p+3)\times 2(p+3)} & I_{2(p+3)\times 2(p+3)} & 0_{2(p+3)\times(D-4p-12)} \\ 0 & 0 & 0 \\ 0 & 0 & 0 \end{bmatrix} \in \mathbb{R}^{D\times D},$$

which will induce the output on this layer as

$$M_2 = M_1 + \mathcal{A}ttn(M_1, QK^{(2)}, V^{(2)}) = [[M_0]_{(\cdot,p+3)}, Z_1, Z_2, Z_3, [M_0]_{(\cdot,4(p+3)+1:D)}].$$

Repeating such construction $\mathcal{O}(\ln L)$ times, we can obtain the $(L-1)$-step history (see Figure 5 for a detailed illustration). Now the output matrix should be

$$M_h = [[M_0]_{(\cdot,1:p+3)}, Z_1, Z_2, Z_3, \ldots, Z_{L-1}, [M_0]_{(\cdot,L(p+3)+1:D)}] \in \mathbb{R}^{(n(L+1)+k)\times D}.$$

On the following layer, we consider the Query-Key matrix as

$$QK^{(f)} := \begin{bmatrix} 0_{(p+1)\times(p+1)} & 0 & 0 \\ 0 & B & 0 \\ 0 & 0 & 0 \end{bmatrix}, \quad B := \beta_1 \begin{bmatrix} \cos(\frac{1}{1000nk}) & -\sin(\frac{1}{1000nk}) \\ \sin(\frac{1}{1000nk}) & \cos(\frac{1}{1000nk}) \end{bmatrix},$$

and the value matrix is constructed as

$$V^{(f)} := \begin{bmatrix} 0_{(p+3)\times L(p+3)} & I_{(p+3)\times(p+3)} & 0_{(p+3)\times(D-(L+1)(p+3))} \\ 0 & 0 & 0 \\ 0 & 0 & 0 \end{bmatrix} \in \mathbb{R}^{D\times D},$$

which implies that sending $\beta_1 \to \infty$, the output on each row should be

$$\left[\text{Softmax}\left([M_0]_{(t,\cdot)}QK^{(1)}M_0^T\right)M_0V^{(1)}\right]_{(t,\cdot)} = [0, [M_0]_{(t+1,1:p+3)}, 0]^T, \quad \forall 1 \le t < n(L+1)+k,$$

So the output decouple matrix after this layer should be

$$M_{\text{dec}} = [[M_0]_{(\cdot,p+3)}, Z_1, Z_2, Z_3, \ldots, Z_{L-1}, F_1, [M_0]_{(\cdot,(L+1)(p+3)+1:D)}].$$

Then the decoupled feature learning process has been finished, which needs $\mathcal{O}(\ln L)$ layers (see details in Figure 5).

**Gradient descent performing.** The following $\mathcal{O}(T)$-layer $2p$-head architecture is designed to learn $\hat{\mathbb{P}}_L(\cdot|z_k)$ based on history information $\{Z_1, \ldots, Z_{L-1}\}$. The construction follows immediately from Lemma 7. To be specific, from Eq. (6), we need to take linear regression to estimate a matrix $\hat{W} \in \mathbb{R}^{p\times p(L-1)}$. Based on the $n$ samples collected, the estimation process is based on MSE loss, i.e,

$$\arg\min_W \mathcal{L}(W) := \arg\min_W \sum_i \|o_{i,L} - Wz_i\|_2^2,$$

where $z_i$ refers to the $(L-1)$-step history of $o_{i,L}$, which has been learned in previous layers. To perform such estimation process for $W$, we construct an $2p$-head $\mathcal{O}(T)$-layer Attention. Each layer can perform one step gradient descent on $\mathcal{L}(W)$ with initial value 0, and each row of $W$ is assigned to be learned by two heads independently (see Figure 6 for detailed illustration). Here we take the updating for $[W]_{(1,\cdot)}$ as an example, and denote the initial point as $0_{p(L-1)}$, which has been stored in $[M_{\text{dec}}]_{(t,(L+1)(p+3)+1:(L+1)(p+3)+p(L-1))}$ on each $1 \le t \le n(L+1)+k$. The gradient vector is

$$\partial\mathcal{L}/\partial W_{(1,\cdot)} = 2\sum_i (W_{(1,\cdot)}^T z_i - [o_{i,L}]_1) \cdot z_i$$

$$= 2\sum_i \left(\text{ReLU}(W_{(1,\cdot)}^T z_i - [o_{i,L}]_1) - \text{ReLU}(-W_{(1,\cdot)}^T z_i + [o_{i,L}]_1)\right) \cdot z_i. \tag{7}$$

The construction will show that each attention layer is related to one-step gradient descent with learning rate $(L-1)^{-1}$, and the construction for each layer is the same. As the first two heads on each layer is related to the updating for $[W]_{(1,\cdot)}$, we design the first Attention head on each layer with Query-Key matrix as

$$
\left([M_{\text{dec}}]_{(t_1,\cdot)}Q^{(g,1)}\right)^T = \begin{bmatrix} [W]_{(1,\cdot)} \\ -1 \\ -\beta_2 1_p \\ 0 \\ -\beta_2 \end{bmatrix}, \quad K^{(g,1)}[M_{\text{dec}}]_{(t_2,\cdot)} = \begin{bmatrix} [Z_1]_{(t_2,1:p)} \\ \cdots \\ [Z_{L-1}]_{(t_2,1:p)} \\ [M_0]_{(t_2,1)} \\ [F_1]_{(t_2,1:p)} \\ 0 \\ 1(t_2 > n(L+1)) \end{bmatrix},
$$

for any $1 \le t_1, t_2 \le n(L+1) + k$. Choosing $\beta_2 > 1000nk$, with ReLU activation function, we obtain

$$
\text{ReLU}\left([M_{\text{dec}}]_{(t_1,\cdot)}^T Q^{(g,1)} K^{(g,1)}[M_{\text{dec}}]_{(t_2,\cdot)}\right) = \begin{cases} \text{ReLU}\left([W]_{(1,\cdot)}^\top z'_{t_2} - [M_0]_{(t_2,1)}\right), & [F_1]_{(t_2,1:p+1)} = o_{\text{delim}}, \\ 0, & \text{otherwise}, \end{cases}
$$

where we denote $z'_t := [[Z_1]_{(t_2,1:p)}^T, \dots, [Z_{L-1}]_{(t_2,1:p)}^T]^T \in \mathbb{R}^{p(L-1)}$. Then with the Value matrix satisfying that

$$
V^{(g,1)}[M_{\text{dec}}]_{(t_2,\cdot)}^T = \frac{1}{L-1} \begin{bmatrix} 0 \\ [Z_1]_{(t_2,1:p)} \\ \cdots \\ [Z_{L-1}]_{(t_2,1:p)} \\ 0 \end{bmatrix},
$$

we can obtain the value on each row of the output matrix:

$$
\left[\mathcal{A}ttn\left(M_{\text{dec}}, Q^{(g,1)}, K^{(g,1)}, V^{(g,1)}\right)\right]_{(t,\cdot)} = \left[0, \frac{1}{L-1}\sum_i \text{ReLU}\left([W]_{(1,\cdot)}^\top z_i - [o_{i,L}]_1\right), 0\right],
$$

for any $1 \le t \le n(L+1) + k$. Also, we consider another Attention head for $W_{1,\cdot}$ with $\{-Q^{(q,1)}, K^{(g,1)}, V_{(g,1)}\}$, the output on each row should be

$$
\left[\mathcal{A}ttn\left(M_{\text{dec}}, -Q^{(g,1)}, K^{(g,1)}, -V^{(g,1)}\right)\right]_{t,\cdot} = \left[0, -\frac{1}{L-1}\sum_i \text{ReLU}\left(-[W]_{(1,\cdot)}^\top z_i + [o_{i,L}]_1\right), 0\right].
$$

Taking summation on both of the two heads, we can finish the update on $[W]_{(1,\cdot)}$ as in Eq. (7). The updates on other rows of $W$ are similar, so with such $2p$ Attention heads on each layer, we can finish one-step gradient descent on MSE loss by

$$
M_{\text{dec}} + \sum_{j=1}^p \mathcal{A}ttn\left(M_{\text{dec}}, Q^{(g,j)}, K^{(g,j)}, V^{(g,j)}\right) + \mathcal{A}ttn\left(M_{\text{dec}}, -Q^{(g,j)}, K^{(g,j)}, -V^{(g,j)}\right).
$$

Considering $\mathcal{O}(T)$ layers with the same structure, we can obtain $\hat{W}$ with a small error. Now the output matrix should be

$$
M_{\text{gd}} = [[M_0]_{(\cdot,p+3)}, Z_1, Z_2, Z_3, \dots, Z_{L-1}, F_1, [W]_{(1,\cdot)}, \dots, [W]_{(p,\cdot)}, [M_0]_{(\cdot,(L+1)(p+3)+p^2(L-1)+1:D)}].
$$

**Prediction with decoupled features.** Finally, on the last layer, we construct a $2p$-head Attention to make prediction on $\hat{\mathbb{P}}_L(\cdot|o_{\text{test},k-1}, \dots, o_{\text{test},k-L+1})$, and each dimension is corresponding to two Attention heads. To be specific, for the first dimension of $\hat{\mathbb{P}}_L(\cdot|o_{\text{test},k-1}, \dots, o_{\text{test},k-L})$, Attention head is designed with

$$
\left([M_{\text{gd}}]_{(t_1,\cdot)}Q^{(pre,1)}\right)^T = \begin{bmatrix} [Z_1]_{(t_2,1:p)} \\ \cdots \\ [Z_{L-1}]_{(t_2,1:p)} \\ 0 \end{bmatrix}, \quad K^{(pre,1)}[M_{\text{gd}}]_{(t_2,\cdot)}^T = \begin{bmatrix} [W]_{(1,\cdot)} \\ 0 \end{bmatrix},
$$

$$
V^{(pre,1)}[M_{\text{gd}}]_{(t_2,\cdot)}^T = \begin{bmatrix} \frac{1}{n(L+1)+k} \\ 0 \end{bmatrix}.
$$

Then we will obtain

$$\left[\mathcal{A}ttn\left(M_{\mathrm{gd}}, Q^{(pre,1)}, K^{(pre,1)}, V^{(pre,1)}\right)\right]_{(n(L+1)+k,\cdot)} = \left[\mathrm{ReLU}\left([W]_{(1,\cdot)}^{\top} o_{\mathrm{test},k-L+1:k-1}\right), 0\right],$$

and

$$\left[\mathcal{A}ttn\left(M_{\mathrm{gd}}, Q^{(pre,1)}, K^{(pre,1)}, V^{(pre,1)}\right) + \mathcal{A}ttn\left(M_{\mathrm{gd}}, -Q^{(pre,1)}, K^{(pre,1)}, -V^{(pre,1)}\right)\right]_{(n(L+1)+k,\cdot)}$$

$$= \left[[W]_{(1,\cdot)}^{\top} o_{\mathrm{test},k-L+1:k-1}, 0\right],$$

which finish the prediction on $\hat{\mathbb{P}}_L(o_{\mathrm{test},k} = e_1 | o_{\mathrm{test},k-1}, \ldots, o_{\mathrm{test},k-L})$. The constructions on other $2p - 2$ heads are similar.

**Optimization error.** Then we turn to the approximation for $\epsilon_3$, which is induced by the finite gradient steps ($\mathcal{O}(T)$ steps) the transformer performs. The error could be estimated directly from Lemma 7. Denoting

$$Z = [o_{1,1:L-1}, \ldots, o_{n,1:L-1}] \in \mathbb{R}^{p(L-1)\times n},$$

from Assumption 3, we have

$$\alpha \le \lambda_{\min}\left(\frac{1}{n}ZZ^T\right) \le \lambda_{\max}\left(\frac{1}{n}ZZ^T\right) \le L, \quad \|o_{\mathrm{test},k-L+1:k-1}\|_2 = \sqrt{L-1}, \quad \|[W_*]_{(j,\cdot)}\|_2 = \mathcal{O}(1),$$

so

$$\epsilon_3 = \mathcal{O}\left(e^{-\alpha T/(2L)} pL^{1/2} \max_{j\in[p]} \|[W_*]_{(j,\cdot)}\|_2\right) = \mathcal{O}(pL^{1/2} e^{-\alpha T/(2L)}).$$

### D.2 GENERALIZATION ERROR

For $\epsilon_2$, we can express the solution $\hat{W}$ for Eq. (6) as

$$\hat{W} := OZ^T(ZZ^T)^{-1},$$

where we use the notation

$$O := \begin{bmatrix} o_{1,L} & o_{2,L} & \cdots & o_{n,L} \end{bmatrix} \in \mathbb{R}^{p\times n}, \quad Z = [o_{1,1:L-1}, \ldots, o_{n,1:L-1}] \in \mathbb{R}^{p(L-1)\times n}.$$

Denoting $z_{\mathrm{test}} := o_{\mathrm{test},k-L+1:k-1}$ and $\Delta := O - W_* Z$, we have

$$\epsilon_2 = \mathbb{E}_{o_{\mathrm{test},1:k-1}} \|\mathbb{P}_L(\cdot | o_{\mathrm{test},k-L+1:k-1}) - \hat{\mathbb{P}}_L(\cdot | o_{\mathrm{test},k-L+1:k-1})\|_1$$

$$= \sum_{j=1}^{p} \mathbb{E}_{z_{\mathrm{test}}} \left|([W_*]_{(j,\cdot)}^T - [O]_{(j,\cdot)}^T Z^T(ZZ^T)^{-1}) z_{\mathrm{test}}\right|$$

$$\le \sum_{j=1}^{p} \sqrt{L} \|[W_*]_{(j,\cdot)} - [O]_{(j,\cdot)} Z^T(ZZ^T)^{-1}\|_2$$

$$= \sum_{j=1}^{p} \sqrt{L} \|[W_*]_{(j,\cdot)} - \left([W_*]_{(j,\cdot)} Z + [\Delta]_{(j,\cdot)}\right) Z^T(ZZ^T)^{-1}\|_2$$

$$= \sqrt{L} \sum_{j=1}^{p} \|[\Delta]_{(j,\cdot)} Z^T(ZZ^T)^{-1}\|_2$$

$$\le \frac{\sqrt{L}}{n\alpha} \sum_{j=1}^{p} \|([\Delta]_{(j,\cdot)} - \mathbb{E}_i[[\Delta]_{(j,\cdot)}] + \mathbb{E}_i[[\Delta]_{(j,\cdot)}]) Z^T\|_2$$

$$\le \frac{\sqrt{L}}{n\alpha} \sum_{j=1}^{p} \|([\Delta]_{(j,\cdot)} - \mathbb{E}[[\Delta]_{(j,\cdot)}]) Z^T\|_2 + \frac{\sqrt{L}}{n\alpha} \sum_{j=1}^{p} \|\mathbb{E}[[\Delta]_{(j,\cdot)}]) Z^T\|_2 \qquad (8)$$

where the first inequality uses the Cauchy-Schwartz inequality, and the second inequality is from Assumption 3, where the expectation $\mathbb{E}[[\Delta]_{(j,i)}] = \mathbb{E}_{o_{i,1:k-1}}[\mathbb{P}(e_j \mid o_{1:k-1}) - \mathbb{P}_L(e_j \mid o_{k-L+1:k-1})]$ due to the decomposition:

$$
\begin{aligned}
[\Delta]_{(j,i)} =& [O]_{(j,i)} - [W_*]_{(j,\cdot)} o_{i,1:L-1} \\
=& 1(o_{i,L} = e_j) - \mathbb{P}(e_j|o_{i,1:L-1}) + \mathbb{P}(e_j|o_{i,1:L-1}) - \mathbb{P}_L(e_j|o_{i,1:L-1}).
\end{aligned}
$$

Hence, we can deal with the second term above:

$$
\begin{aligned}
\frac{\sqrt{L}}{n\alpha} \sum_{j=1}^{p} \|\mathbb{E}[[\Delta]_{(j,\cdot)}]) Z^T\|_2 \leq& \frac{L}{n\alpha} \sum_{j=1}^{p} \sum_{i=1}^{n} \mathbb{E}_{o_{i,1:k-1}} |\mathbb{P}(e_j \mid o_{1:k-1}) - \mathbb{P}_L(e_j \mid o_{k-L+1:k-1})| \\
=& \frac{L}{n\alpha} \sum_{i=1}^{n} \mathbb{E}_{o_{i,1:k-1}} \|\mathbb{P}(\cdot \mid o_{i,1:k-1}) - \mathbb{P}_L(\cdot \mid o_{i,k-L+1:k-1})\|_1 \\
\leq& \mathcal{O}\left(\frac{Ld}{\alpha} \cdot e^{-L\gamma^4}\right),
\end{aligned}
$$

where the first inequality uses the formulation that $\|[Z]_{(i,\cdot)}\|_2 \leq \sqrt{L}$, and the second inequality uses Lemma 1.

Next, for the first term in (8), we can define the error $\delta_{j,i} := [\Delta]_{(j,i)} - \mathbb{E}[[\Delta]_{(j,i)}]$. For each $i, j$, $\delta_{j,i}$ is a zero-mean 1-sub-Gaussian variable. We also have for each $i$, $\max\{\|z_i z_i^\top\|_2, \|z_i^\top z_i\|_2\} \leq L$. Thus, we can invoke Lemma 8 to obtain that with probability at least $1 - \frac{1}{n}$, for any $j = 1, \ldots, p$,

$$
\|\left([\Delta]_{(j,\cdot)} - \mathbb{E}[[\Delta]_{(j,\cdot)}]\right) Z^T\|_2 = \|\sum_{i=1}^{n} \delta_{i,j} z_i\|_2 \leq 4\sqrt{nL \ln(2nLp^2)}.
$$

Therefore, by taking the results above back into (8), we can obtain that

$$
\epsilon_2 \leq \mathcal{O}\left(\frac{pL\sqrt{\ln(nLp)}}{\sqrt{n}\alpha} + \frac{Ld}{\alpha} \cdot e^{-L\gamma^4}\right).
$$

# E    PROOF SKETCHES FOR THEOREM 2

We also decompose the prediction error into three parts as in (4) and analyze them correspondingly.

## E.1    MODEL APPROXIMATION

For the model approximation error $\epsilon_1$, under Assumption 4, we can also approximate the $m$-step transition probability $\mathbb{P}(o_{k:k+m} \mid o_{1:k-1})$ by a $(L-1)$-memory probability $\hat{\mathbb{P}}_L(o_{k:k+m} \mid o_{k-L+1:k-1})$. Since we can take $o_{k:k+m}$ as a whole vector, with similar techniques in Section 4.1, we can show that

**Lemma 2.** *For any $\epsilon > 0$, there exists a $\mathcal{O}(L)$-memory transition probability $\hat{\mathbb{P}}_L$ with $L = \Theta(\gamma^{-4} \log(d/\epsilon))$ such that*

$$
\mathbb{E}_{o_{1:k}} \|\mathbb{P}(o_{k:k+m} \mid o_{1:k}) - \mathbb{P}_L(o_{k:k+m} \mid o_{t-L:t})\|_1 \leq \mathcal{O}\left(de^{-L\gamma^4}\right).
$$

This model approximation bound is the same to Lemma 1, and the $\mathbb{P}_L$ also enjoys the low-rank structure

$$
\mathbb{P}_L(o_{k:k+m} \mid o_{k-L+m:k-1}) := \mu(o_{k:k+m})^\top \phi(o_{k-L+m:k-1}),
$$

where $\mu(o_{k:k+m}), \phi(o_{k-L+m:k-1}) \in \mathbb{R}^d$ are representation vectors. For conciseness, we defer the details to Appendix G.

After embedding the $m$-step observation $o_{k:k+m}$ as one-hot vector $\mathcal{V}ec(o_{k:k+m}) \in \mathbb{R}^{p^m}$, we can express the mapping function $\mu(\cdot)$ as

$$
\mu(o_{k:k+m}) = U' \mathcal{V}ec(o_{k:k+m}),
$$

where $U' \in \mathbb{R}^{d \times p^m}$. Considering the linear assumption on $\phi$, similar to Eq. (5), we can also obtain

$$\mathbb{P}_L(\cdot|o_{k-L+m:k-1}) := W'_* o_{k-L+m:k-1},$$

for some $W'_* \in \mathbb{R}^{p^m \times p(L-m)}$. Taking decomposition for the approximation error, we have

$$\mathbb{E}_{o_{\text{test},1:k-1}} \|\mathbb{P}(o_{\text{test},k:k+m-1}|o_{\text{test},1:k-1}) - \text{read}(\text{TF}_\theta(M_0))\|_1$$

$$\leq \underbrace{\mathbb{E}_{o_{\text{test},1:k-1}} \|\mathbb{P}(o_{\text{test},k:k+m-1}|o_{\text{test},1:k-1}) - \mathbb{P}_L(o_{\text{test},k:k+m-1}|o_{\text{test},k-L+m:k-1})\|_1}_{\epsilon_1 : \text{model approximation}}$$

$$+ \underbrace{\mathbb{E}_{o_{\text{test},1:k-1}} \|\mathbb{P}_L(o_{\text{test},k:k+m-1}|o_{\text{test},k-L+m:k-1}) - \hat{\mathbb{P}}_L(o_{\text{test},k:k+m-1}|o_{\text{test},k-L+m:k-1})\|_1}_{\epsilon_2 : \text{generalization}}$$

$$+ \underbrace{\mathbb{E}_{o_{\text{test},1:k-1}} \|\hat{\mathbb{P}}_L(o_{\text{test},k:k+m-1}|o_{\text{test},k-L+m:k-1}) - \text{read}(\text{TF}_\theta(M_0))\|_1}_{\epsilon_3 : \text{optimization}},$$

where $\hat{\mathbb{P}}_L(\cdot|o_{\text{test},k-L+1:k-1})$ refers to the solution based on $n$ samples we collected:

$$\hat{\mathbb{P}}_L(\cdot|o_{\text{test},k-L+m:k-1}) = \hat{W}'[o_{\text{test},k-L+m}, \ldots, o_{\text{test},k-1}]^T,$$

$$\hat{W}' := \arg\min_W \sum_i \|\mathcal{V}ec(o_{i,L-m+1:L}) - W o_{i,1:L-m}\|_2^2.$$

In the error decomposition, $\epsilon_1 = \mathcal{O}(de^{-\gamma^4 L})$ can be obtained from Lemma 2 immediately. And in further analysis, we will estimate $\epsilon_2$ and $\epsilon_3$ respectively.

### E.2 TRANSFORMER CONSTRUCTION

Then the construction is similar to the construction for Theorem 1. So here we just provide a sketch for it.

**Decoupled feature learning.** Recalling the matrix:

$$A := \beta_1 \begin{bmatrix} \cos(\frac{1}{1000nk}) & \sin(\frac{1}{1000nk}) \\ -\sin(\frac{1}{1000nk}) & \cos(\frac{1}{1000nk}) \end{bmatrix}, \quad B := \beta_1 \begin{bmatrix} \cos(\frac{1}{1000nk}) & -\sin(\frac{1}{1000nk}) \\ \sin(\frac{1}{1000nk}) & \cos(\frac{1}{1000nk}) \end{bmatrix},$$

on each time index $t$, we can use $A$ to capture the history information $Z_r$, and use $B$ to capture the future information $F_r$. So with $\mathcal{O}(\ln(L-m) + \ln m) = \mathcal{O}(\ln L)$ layers, we can obtain the output matrix as

$$M_{\text{dec}} = [[M_0]_{(\cdot,1:p+3)}, Z_1, Z_2, \ldots, Z_{L-m}, F_1, F_2, \ldots, F_m, [M_0]_{(\cdot,(L+1)(p+3)+1:D)}].$$

Then before taking gradient descent, we use the one-hot mapping function $\mathcal{V}ec$ on each row of $\{[M_0]_{(\cdot,1:p)}, [F_1]_{(\cdot,1:p)}, \ldots, [F_{m-1}]_{(\cdot,1:p)}\}$, which refers to the current and future observations on each time index. After that, we will obtain

$$M_v := [[M_0]_{\cdot,1:p+3}, Z_1, Z_2, \ldots, Z_{L-m}, F_1, F_2, \ldots, F_m, H, [M_0]_{(\cdot,(L+1)(p+3)+p^m+1:D)}],$$

where

$$[H]_{(t,\cdot)} = \mathcal{V}ec \left[[M_0]_{(t,1:p)}, [F_1]_{(t,1:p)}, \ldots, [F_{m-1}]_{(t,1:p)}\right]^T$$

for each $1 \leq t \leq nL + n + k$.

**Gradient descent and final prediction.** After obtaining these features, we shall perform gradient descent on MSE loss

$$\arg\min_{W'} \sum_i \|\mathcal{V}ec(o_{i,L-m+1:L}) - W' o_{i,1:L-m}\|_2^2.$$

Then we could use $2p^m$-head $\mathcal{O}(T)$-layer Attention to perform the gradient descent on $W$, in which the feature $H$ and $\{Z_1, \ldots, Z_{L-m}\}$ will be taken into consideration. The construction is similar to Theorem 1.

**Optimization error.** For $\epsilon_3$, under Assumption 3, we can also use Lemma 5 to obtain that

$$\epsilon_3 = \mathcal{O}\left(p^m L^{1/2} e^{-\alpha T/(2L)}\right).$$

### E.3 GENERALIZATION ERROR

We can rewrite $\hat{\mathbb{P}}_L(\cdot|o_{\text{test},k-L+1:k-m})$ as

$$\hat{\mathbb{P}}_L(\cdot|o_{\text{test},k-L+m:k-1}) = \hat{W}' o_{\text{test},k-L+m:k-1}, \quad \hat{W}' = O_m Z_m^T (Z_m Z_m^T)^{-1},$$

where we denote

$$O_m := [\mathcal{V}ec(o_{1,L-m+1:L}) \quad \mathcal{V}ec(o_{2,L-m+1:L}) \quad \cdots \quad \mathcal{V}ec(o_{n,L-m+1:L})] \in \mathbb{R}^{p^m \times n},$$

$$Z_m := [o_{1,1:L-m} \quad o_{2,1:L-m} \quad \cdots \quad o_{n,1:L-m}] \in \mathbb{R}^{(L-m) \times n}.$$

Denoting $z_{\text{test}} := o_{\text{test},k-L+m:k-1}$ and $\Delta := O_m - W_*' Z_m$, we have

$$\epsilon_2 = \mathbb{E}_{o_{\text{test},1:k-1}} \|\mathbb{P}_L(\cdot|o_{\text{test},k-L+m:k-1}) - \hat{\mathbb{P}}_L(\cdot|o_{\text{test},k-L+m:k-1})\|_1$$

$$= \sum_{j=1}^{p^m} \mathbb{E}_{z_{\text{test}}} \left| ([W_*']_{(j,\cdot)}^T - [O_m]_{(j,\cdot)}^T Z_m^T (Z_m Z_m^T)^{-1}) z_{\text{test}} \right|$$

$$\leq \sum_{j=1}^{p^m} \sqrt{L} \|[W_*']_{(j,\cdot)} - [O_m]_{(j,\cdot)} Z_m^T (Z_m Z_m^T)^{-1}\|_2$$

$$= \sum_{j=1}^{p^m} \sqrt{L} \|[W_*']_{(j,\cdot)} - ([W_*']_{(j,\cdot)} Z_m + [\Delta]_{(j,\cdot)}) Z_m^T (Z_m Z_m^T)^{-1}\|_2$$

$$= \sqrt{L} \sum_{j=1}^{p^m} \|[\Delta]_{(j,\cdot)} Z_m^T (Z_m Z_m^T)^{-1}\|_2$$

$$\leq \frac{\sqrt{L}}{n\alpha} \sum_{j=1}^{p^m} \|([\Delta]_{(j,\cdot)} - \mathbb{E}_i[[\Delta]_{(j,\cdot)}] + \mathbb{E}_i[[\Delta]_{(j,\cdot)}]) Z_m^T\|_2$$

$$\leq \frac{\sqrt{L}}{n\alpha} \sum_{j=1}^{p^m} \|([\Delta]_{(j,\cdot)} - \mathbb{E}[[\Delta]_{(j,\cdot)}]) Z_m^T\|_2 + \frac{\sqrt{L}}{n\alpha} \sum_{j=1}^{p^m} \|\mathbb{E}[[\Delta]_{(j,\cdot)}]) Z_m^T\|_2, \quad (9)$$

where the first inequality uses the Cauchy-Schwartz inequality, and the second inequality is from Assumption 3, where the expectation $\mathbb{E}[[\Delta]_{(j,i)}] = \mathbb{E}_{o_{i,1:k-1}}[\mathbb{P}(e_j \mid o_{1:k-1}) - \mathbb{P}_L(e_j \mid o_{k-L+m:k-1})]$ due to the decomposition:

$$[\Delta]_{(j,i)} = [O]_{(j,i)} - [W_*']_{(j,\cdot)} o_{i,1:L-m}$$

$$= 1(o_{i,L-m+1:L} = e_j) - \mathbb{P}(e_j|o_{i,1:L-m}) + \mathbb{P}(e_j|o_{i,1:L-m}) - \mathbb{P}_L(e_j|o_{i,1:L-m}).$$

Hence, we can deal with the second term above:

$$\frac{\sqrt{L}}{n\alpha} \sum_{j=1}^{p^m} \|\mathbb{E}[[\Delta]_{(j,\cdot)}]) Z_m^T\|_2 \leq \frac{L}{n\alpha} \sum_{j=1}^{p^m} \sum_{i=1}^n \mathbb{E}_{o_{i,1:k-1}} |\mathbb{P}(e_j \mid o_{1:k-1}) - \mathbb{P}_L(e_j \mid o_{k-L+m:k-1})|$$

$$= \frac{L}{n\alpha} \sum_{i=1}^n \mathbb{E}_{o_{i,1:k-1}} \|\mathbb{P}(\cdot \mid o_{i,1:k-1}) - \mathbb{P}_L(\cdot \mid o_{i,k-L+m:k-1})\|_1$$

$$\leq \mathcal{O}(\frac{Ld}{\alpha} \cdot e^{-L\gamma^4}),$$

where the first inequality uses the formulation that $\|[Z_m]_{(i,)}\|_2 \leq \sqrt{L}$, and the second inequality uses Lemma 2.

Next, for the first term in (9), we can define the error $\delta_{j,i} := [\Delta]_{(j,i)} - \mathbb{E}[[\Delta]_{(j,i)}]$. For each $i,j$, $\delta_{j,i}$ is a zero-mean 1-sub-Gaussian variable. We also have for each $i$,

$\max\{\|[Z_m]_{(\cdot,i)}[Z_m]_{(\cdot,i)}^\top\|_2, \|[Z_m]_{(\cdot,i)}^\top[Z_m]_{(\cdot,i)}\|_2\} \le L$. Thus, we can invoke Lemma 8 to obtain that with probability at least $1 - \frac{1}{n}$, for any $j = 1, \ldots, p^m$,

$$\|\big([\Delta]_{(j,\cdot)} - \mathbb{E}[[\Delta]_{(j,\cdot)}]\big)Z^T\|_2 = \|\sum_{i=1}^n \delta_{i,j}[Z_m]_{(\cdot,i)}\|_2 \le 4\sqrt{nL\ln(2nLp^{m+1})}.$$

Therefore, by taking the results above back into (9), we can obtain that

$$\epsilon_2 \le \mathcal{O}\Big(\frac{p^m L\sqrt{\ln(nLp)}}{\sqrt{n}\alpha} + \frac{Ld}{\alpha} \cdot e^{-L\gamma^4}\Big).$$

## F  PROOF FOR LEMMA 1

To facilitate analysis, we define the belief state $b_k(o_{1:k-1}) \in \Delta(\mathcal{H})$ as the posterior given observations: $b_k(o_{1:k})(h) = \mathbb{P}(h_k \mid o_{1:k})$. Combining this notation and the low-rank hidden-state transition, we can write

$$\mathbb{P}(o_k \mid o_{1:k-1}) = \sum_{h_k, h_{k-1}} \mathbb{P}(o_k \mid h_k)\mathbb{P}(h_k \mid h_{k-1})\mathbb{P}(h_{k-1} \mid o_{1:k-1})$$

$$= \Big(\sum_{h_k} \mathbb{T}(o_k \mid h_k)w^*(h_k)\Big)^\top \cdot \Big(\sum_{h_{k-1}} \psi^*(h_{k-1})b(o_{1:k-1})(h_{k-1})\Big).$$

The transition is the inner product of $d$-dimensional representations of history $o_{1:k-1}$ and next token $o_k$. Especially, the historical information is embedded into the belief state. Thus, to approximate $\mathbb{P}$ by $\mathbb{P}_L$, we need to approximate $b(o_{1:k-1})$ by a $(L-1)$-memory belief state $b_L(o_{k-L+1:k-1})$. Assumption 4 implies that we can reverse the inequality to obtain the contraction from observation to hidden state distributions

$$\|d - d'\|_1 \le \gamma^{-1}\|\mathbb{T}d - \mathbb{T}d'\|_1.$$

Hence, by constructing a history-independent belief state $\tilde{b}_0$ within a KL-ball of $b$: $\mathrm{KL}(b, \tilde{b}_0) \le d^3$ (which can be realized by G-optimal design), the belief state $b_L(o_{k-L+1:k-1})$ induced from $\tilde{b}_0$ can gradually approximate $b(o_{1:k-1})$ that has the same $(L-1)$-length observations. Theorem 14 of Uehara et al. (2022) demonstrated that

**Lemma 3** (Theorem 14 of Uehara et al. (2022))**.** *Under Assumption 4, for $K \ge L + 1$, $L \ge C\gamma^{-4}\log(d/\epsilon)$, where $C > 0$ is a constant, we have*

$$\mathbb{E}_{o_{1:k-1}}\big\|b(o_{1:k-1}) - b_L(o_{k-L+1:k-1})\big\|_1 \le \epsilon. \tag{10}$$

*Proof of Lemma 1.* The proof is the same to Proposition 7 of Guo et al. (2023a). The only difference is there is no actions in HMM. For any $k \ge L + 1$, given the $b_L$ satisfying (10), now, we can construct the probability as

$$\mathbb{P}_L(o_k \mid o_{k-L+1:k-1}) = \Big(\sum_{h_k} \mathbb{T}(o_k \mid h_k)w^*(h_k)\Big)^\top \cdot \Big(\sum_{h_{k-1}} \psi^*(h_{k-1})b_L(o_{k-L+1:k-1})(h_{k-1})\Big)$$

$$:= \mu(o_k)^\top \phi(o_{k-L+1:k-1}),$$

where we use the notation

$$\mu(o_k) = \sum_{h_k} \mathbb{T}(o_k \mid h_k)w^*(h_k), \quad \phi(o_{k-L+1:k-1}) = \sum_{h_{k-1}} \psi^*(h_{k-1})b_L(o_{k-L+1:k-1})(h_{k-1}).$$

Hence, we deduce that

$$\mathbb{E}_{o_{1:k-1}}\mathbb{P}(o_k \mid o_{1:k-1}) = \mathbb{E}_{o_{1:k-1}} \sum_{h_k} \mathbb{T}(o_k \mid h_k) w^*(h_k)^\top \cdot \sum_{h_{k-1}} \psi^*(h_{k-1}) b(o_{1:k-1})(h_{k-1})$$

$$\leq \mathbb{E}_{o_{1:k-1}} \sum_{h_k} \mathbb{T}(o_k \mid h_k) w^*(h_k)^\top$$

$$\cdot \sum_{h_{k-1}} \psi^*(h_{k-1}) \Big( \big| b(o_{1:k-1})(h_{k-1}) - b_L(o_{k-L+1:k-1})(h_{k-1}) \big| + b_L(o_{k-L+1:k-1})(h_{k-1}) \Big)$$

$$= \mathbb{E}_{o_{1:k-1}} \sum_{h_k} \mathbb{T}(o_k \mid h_k) w^*(h_k)^\top \cdot \sum_{h_{k-1}} \psi^*(h_{k-1}) b_L(o_{k-L+1:k-1})(h_{k-1})$$

$$+ \mathbb{E}_{o_{1:k-1}} \sum_{h_k} \mathbb{T}(o_k \mid h_k) w^*(h_k)^\top \cdot \sum_{h_{k-1}} \psi^*(h_{k-1}) \big| b(o_{1:k-1})(h_{k-1}) - b_L(o_{k-L+1:k-1})(h_{k-1}) \big|.$$

$$(11)$$

Since we have for any $h_{k-1}$

$$\sum_{h_k} \mathbb{T}(o_k \mid h_k) w^*(h_k)^\top \psi^*(h_{k-1}) = \sum_{h_k} \mathbb{P}(o_k \mid h_k) \mathbb{P}(h_k \mid h_{k-1}) \leq 1,$$

term (11) can be bounded as

$$\mathbb{E}_{o_{1:k-1}} \sum_{h_{k-1}} \Big( \sum_{h_k} \mathbb{T}(o_k \mid h_k) w^*(h_k)^\top \cdot \psi^*(h_{k-1}) \Big) \cdot \big| b(o_{1:k-1})(h_{k-1}) - b_L(o_{k-L+1:k-1})(h_{k-1}) \big|$$

$$\leq \mathbb{E}_{o_{1:k-1}} \big| b(o_{1:k-1})(h_{k-1}) - b_L(o_{k-L+1:k-1})(h_{k-1}) \big|$$

$$\leq \epsilon,$$

where the first inequality is by the Cauchy-Schwarz inequality, and the second inequality uses Lemma 4. Therefore, we obtain

$$\mathbb{E}_{o_{1:k-1}}\mathbb{P}(o_k \mid o_{1:k-1}) \leq \mathbb{E}_{o_{1:k-1}}\mathbb{P}_L(o_k \mid o_{k-L+1:k-1}) + \epsilon,$$

which concludes the proof. $\qquad\square$

Then, we can construct the $(L-1)$-memory probability by replacing the belief state

$$\mathbb{P}_L(o_k \mid o_{k-L+1:k-1}) = \Big( \sum_{h_k} \mathbb{T}(o_k \mid h_k) w^*(h_k) \Big)^\top \cdot \Big( \sum_{h_{k-1}} \psi^*(h_{k-1}) b_L(o_{k-L+1:k-1})(h_{k-1}) \Big)$$

$$:= \mu(o_k)^\top \phi(o_{k-L+1:k-1}),$$

## G    PROOF FOR LEMMA 2

*Proof of Lemma 2.* Under the operator $\mathbb{M}$, we can write

$$\mathbb{P}(o_{k:k+m} \mid h_t) = \int_{\mathcal{H}} \mathbb{M}(o_{k:k+m} \mid h_{t+1}) w^*(h_{t+1})^\top \psi^*(h_t) \mathrm{d}h_t.$$

We wish to approximate

$$\mathbb{P}(o_{k:k+m} \mid o_{1:h}) \text{ by } \mathbb{P}_L(o_{k:k+m} \mid o_{t-L+1:t}).$$

Given a history observation $o_{1:k}$, we define the belief state $b_t(o_{1:k}) \in \Delta(\mathcal{S})$ as the distribution

$$b_t(o_{1:k})(h) = \mathbb{P}(h_k = h \mid o_{1:k}).$$

Additionally, for any distribution $b \in \Delta(\mathcal{S})$, we define the belief update operator $B_{k-1}(b, o_{k:k+m})$ as

$$B_{k-1}(b, o_{k:k+m})(h) = \frac{\mathbb{M}(o_{k:k+m} \mid h) \sum_{h'} b(h') \mathbb{P}(h|h')}{\sum_{h''} \mathbb{M}(o_{k:k+m} \mid h'') \sum_{h'} b(h') \mathbb{P}(h''|h')}.$$

then, the update for belief state is

$$b(o_{1:k-1}) = B(b_{k-1}(o_{1:k-1}), o_{k:k+m}).$$

Given this notation, we can write $\mathbb{P}$ as

$$\mathbb{P}(o_{k:k+m} \mid o_{1:k-1}) = \Big( \int_{\mathcal{H}} \mathbb{M}(o_{k:k+m} \mid h_{t+1}) w^*(h_{t+1}) \mathrm{d}h_{t+1} \Big)^\top \cdot \int_{\mathcal{H}} \psi^*(h_{k-1}) b(o_{1:k-1})(h_{k-1}) \mathrm{d}h_{k-1}.$$
(12)

Thus, to approximate $\mathbb{P}$ by $\mathbb{P}_L$, it suffices to approximate $b(o_{1:k})$ by some belief state $b_L(o_{t-L:t})$.

To construct a good approximation, we can first construct a history-independent belief distribution $\tilde{b}_0 \in \Delta(\mathcal{S})$ by G-optimal design (Uehara et al., 2022) such that for any belief state

$$\mathrm{KL}(b, \tilde{b}_0) \leq \ln d^3.$$
(13)

**Lemma 4** (Exponential Stability for Low-rank Transition). *Under Assumption 4, for $L \geq C\gamma^{-4} \log(d/\epsilon)$, we have*

$$\mathbb{E}\big\|b(o_{1:k-1}) - b_L(o_{t:t+L})\big\|_1 \leq \epsilon.$$

Then, by following the same analysis as the proof of Lemma 1, we can prove the desired result. $\square$

## H  TECHNICAL LEMMAS

**Lemma 5** (Convergence rate in gradient descent). *Suppose $L$ is $\alpha$-strongly convex and $\beta$-smooth for some $0 < \alpha < \beta$. Then the gradient descent iterates $w_{GD}^{t+1} := w_{GD}^t - \eta \nabla L(w_{GD}^t)$ with learning rate $\eta = \beta^{-1}$ and initialization $w_{GD}^0$ satisfies*

$$\|w_{GD}^t - w^*\|_2^2 \leq e^{-t/\kappa} \cdot \|w_{GD}^0 - w^*\|_2^2,$$

$$L(w_{GD}^t) - L(w^*) \leq \frac{\beta}{2} e^{-t/\kappa} \cdot \|w_{GD}^0 - w^*\|_2^2,$$

*where $\kappa = \beta/\alpha$ is the condition number, and $w^* = \arg\min L(w)$ is the optimizer of function $L - 1$.*

**Lemma 6** (Lemma G.2 in Ye et al. (2023), Theorem 2.29 in Zhang (2023)). *Let $\{\epsilon_t\}$ be a sequence of zero-mean conditional $\sigma$-subGaussian random variable, i.e, $\ln \mathbb{E}[e^{\lambda \epsilon_i} | \mathcal{S}_{i-1}] \leq \lambda^2 \sigma^2/2$, where $\mathcal{S}_{i-1}$ represents the history data. With probability at least $1 - \delta$, for any $t \geq 1$, we have*

$$\sum_{i=1}^t \epsilon_i^2 \leq 2t\sigma^2 + 3\sigma^2 \ln(1/\delta).$$

**Lemma 7** (Theorem 4 in Bai et al. (2023)). *For any $\lambda \geq 0$, $0 \leq \alpha \leq \beta$ with*

$$\kappa := \frac{\beta + \lambda}{\alpha + \lambda},$$

*$B_w > 0$, and $\varepsilon < \frac{B_x B_w}{2}$, there exists an L-layer attention-only transformer $TF_\theta^0$ with*

$$M = \lceil 2\kappa \log(B_x B_w/(2\varepsilon)) \rceil + 1$$

*(With $R := \max\{B_x B_w, B_y, 1\}$) such that the following holds. On any input data $(D, x_{N+1})$ such that the regression problem is well-conditioned and has a bounded solution:*

$$\alpha \leq \lambda_{\min}(X^\top X/N) \leq \lambda_{\max}(X^\top X/N) \leq \beta,$$

$$\|w_{\mathrm{ridge}}^\lambda\|_2 \leq B_w/2,$$

*$TF_\theta^0$ approximates the prediction $\hat{y}_{N+1}$ as*

$$\big|\hat{y}_{N+1} - \langle w_{\mathrm{ridge}}^\lambda, x_{N+1}\rangle\big| \leq \varepsilon.$$

**Lemma 8** (Lemma F.3 of Fan et al. (2023)). *Consider a sequence of matrix $\{A_t\}_{t=1}^\infty$ with dimension $d_1 \times d_2$ and an i.i.d. sequence $\{\epsilon_t\}_{t=1}^\infty$, where $\epsilon_t$ is conditional $\sigma$-subgaussian (i.e., $\mathbb{E}(e^{\alpha \epsilon_t} | A_t) \leq e^{\alpha^2 \sigma^2/2}$ almost surely for all $\alpha \in \mathbb{R}$). Define the matrix sub-Gaussian series $S = \sum_{t=1}^n \epsilon_t A_t$ with bounded matrix variance statistic:*

$$\max\left\{\big\|A_t A_t^\top\big\|_{op}, \big\|A_t^\top A_t\big\|_{op}\right\} \leq v_t.$$

*Then, for all $u > 0$, we have*

$$\mathbb{P}\left(\|S\|_{op} \geq u\right) \leq (d_1 + d_2) \exp\left(-\frac{u^2}{16\sigma^2 \sum_{t=1}^n v_t}\right).$$

# I LLM USAGE STATEMENT

We used LLMs to aid in polishing the writing of this paper. Specifically, LLMs were employed as a general-purpose assistant to improve clarity, grammar, and style, and to suggest alternative phrasings for technical explanations. They were not used to generate novel research ideas, design experiments, or produce results. The authors take full responsibility for all content, including text refined with the assistance of LLMs.

