# OpenReview forum: "Transformers as Multi-task Learners: Decoupling Features in Hidden Markov Models"
_ICLR.cc/2026/Conference — Submitted to ICLR 2026_

### Official Review · Reviewer_4m3J · 2025-10-30

**Soundness:** 3
**Presentation:** 2
**Contribution:** 3
**Rating:** 6
**Confidence:** 3

**Summary:**

The paper studies how (well) transformers learn Hidden Markov Models (HMMs).  They do so by starting with experimental observations of the same setup, i.e., learning HMMs using transformers, wherein they observe that the state is learned before the overall task.  They use these observations to come up with an architecture that learns transformers and obtain bounds on the expressive power of transformers (in learning HMMs).

**Strengths:**

1.  The paper attempts to provide a strong theoretical foundation to how transformers learn --- they do so by studying how transformers learn an HMM, providing a significant step up in generality than simple Markov chains, which seems to have been standard so far.

2. The theoretical analysis is done correctly and reasonably clearly, there are no mistakes as far as I can see.

**Weaknesses:**

1. The notation is not very clear and this makes the paper rather hard to read, at least on the first pass.  Here are my notation-specific questions, but some suggestions are in Minor comments under Questions below.
      What does \mathcal{O} represent?  My guess is that it represents the observation space, the set from which the observations are drawn.  Then what is \mathcal{O}(\mathcal{H}) (typo, should be \Delta?)?  And what does the "emission operator" mean? What does the * do, and why does the * not appear when \mathbb{T} is referred to again (in, say Assumption 2)? (I could not find any explanation in the paper, but if it is explained somewhere a reference would help.)

2. The natural question is what this tells us about more realistic models.  Theoretical analyses are useful, but the goal is never to keep increasing the complexity of models we study till we can analyze practical models.  The goal is to obtain some useful insights that extend to large models and guide intuition.  It is not clear to me that this paper has any such insights (though the analysis is certainly appreciated).

As such, I recommend acceptance, but not strongly.  I will be happy to increase my score if the authors can convince me that there are potential useful insights to be obtained.  I would also like to ask the authors to improve some of the presentation to make it easier to read (as in the Weakness above and Questions below).

**Questions:**

1. Section 2: Figure 2 is supposed to show that the features become decoupled at deeper layers, but there seems to be some weird phenomena happening at layers 13--14.  Comments?  What exactly is the experiment doing --- what does the random shuffling of positions represent?

2. I appreciate that Assumptions 1, 2, 3 are reasonable analytically and have been made by previous works, but it would be useful to know if these assumptions hold in practice, at least approximately.  Could you comment on this?

3. Why does the low-rank structure in Assumption 1 lead to the "low-rank" transition at line 270 (top of page 6)?  I suspect it might follow from elementary probability, but I was unable to convince myself of this, could you give a proof?

4. Remark 1 seems rather superfluous: it should not be surprising that your explicit construction "aligns closely" with the empirical observations --- if anything, the empirical observations likely helped with the explicit construction?  And then yes, the empirical observation is valid and interesting, and so is the explicit theoretical construction, but the "connection" just seems forced when it is causal, not a correlation.

Minor comments:
1. Section 3.2 is a little hard to read, mainly because of all the notation.  It might be easier to follow if the dimensions of s, v, o are written before they are introduced in the equation with the matrices M_{0,i} and M_{0,test}.
2. Section 3.2: why do you need D > 2p^2 L?  It would be helpful to refer to the part of the analysis where this constraint shows up.  Is this condition normally met in practice?

---

> ### Author Response · Authors · 2025-11-21
> **Response to Reviewer 4m3J(1/2)**
>
> Many thanks for your valuable feedback. Here we make the point-to-point response as follows:
>
> **The notation is not very clear and this makes the paper rather hard to read, at least on the first pass. Here are my notation-specific questions...**
>
> Sorry for the confusing, we will check the notation and revise them. $\mathcal{O}$ refers to the observation space. And there should be $\Delta(\mathcal{H})$. The emission operator means the mapping from the distribution of hidden states to the distribution of observation. In Assumption 2, there should be $\mathbb{T}^*$ instead of $\mathbb{T}$.
>
> **The natural question is what this tells us about more realistic models. Theoretical analyses are useful, but the goal is never to keep increasing the complexity of models we study till we can analyze practical models. The goal is to obtain some useful insights that extend to large models and guide intuition. It is not clear to me that this paper has any such insights (though the analysis is certainly appreciated).**
>
> Thank you for the thoughtful comment. In this work, our primary goal is to clarify and explain the feature-decoupling phenomenon, which we believe may have implications for practical large-scale models. Specifically:
>  - Since the features learned in the lower layers rely mainly on local neighboring tokens, one potential implication is that far-history tokens could be masked (especially for long context lengths), which may improve model efficiency.
>  - Because features become decoupled in higher layers, it may be possible to reduce the number of attention heads in these layers, as fewer heads may be sufficient to learn information.
>  - The decoupling phenomenon also suggests the possibility of more parallelizable architectures in higher layers, which could further improve computational efficiency.
>
> We view these points as promising directions, and we leave the detailed exploration of such practical implications to future work.
>
> **Section 2: Figure 2 is supposed to show that the features become decoupled at deeper layers, but there seems to be some weird phenomena happening at layers 13--14. Comments? What exactly is the experiment doing --- what does the random shuffling of positions represent?**
>
>
> We thank the reviewer for the careful inspection of Figure 2. The metric $$1 - \frac{\text{std}}{\text{mean}}$$ reflects how insensitive each head is to positional perturbations. The slight dip in a few heads around layers 13–14 does not contradict our theory. First, these heads still show higher metrics than most early layers, and the average stability in those layers remains high. Second, our claim does not require that all heads in the deeper layers become fully decoupled, but that decoupled representations emerge predominantly in the upper layers.
>
> **I appreciate that Assumptions 1, 2, 3 are reasonable analytically and have been made by previous works, but it would be useful to know if these assumptions hold in practice, at least approximately. Could you comment on this?**
>
> Thank you for the question. We agree that understanding whether these assumptions hold (even approximately) in practice is important. Assumption 1 is a standard condition in reinforcement learning and is commonly used in prior theoretical analyses (Agarwal et al., 2020; Uehara et al., 2021; 2022; Guo et al., 2023a), which are already cited in the paper. In practice, for example in tabular cases, the transition matrix $P=W\Psi^T$. This condition means the transition matrix is decomposed into two low-rank matrices, and this low-rank assumption holds in lots of scenarios with high-dimensional data, such as Robot Navigation. The environment information is high-dimensional but the state transition is determined by low-dimension latent common factors. Assumption 2 requires that the observations be sufficiently diverse to distinguish hidden states. This aligns with some practical scenarios where meaningful representations allow models to infer latent structure. Moreover, we further relax this requirement in Assumption 4, making the condition milder and more applicable to real-world data. Assumption 3 concerns having a sufficiently large number of sequences nnn. This is also consistent with practice: modern sequence models are typically trained on large datasets, so this assumption is often approximately satisfied.
>
> **Why does the low-rank structure in Assumption 1 lead to the "low-rank" transition at line 270 (top of page 6)? I suspect it might follow from elementary probability, but I was unable to convince myself of this, could you give a proof?**
>
> The proof this equation is provided in equation (14) and (15) in page 40 of [1]. The analysis basically uses the low-rank definition and the integration for conditional probability.
>
> [1] Jiacheng Guo, et. al., Provably Efficient Representation Learning with Tractable Planning in Low-Rank POMDP

---

> ### Author Response · Authors · 2025-11-21
> **Response to Reviewer 4m3J(2/2)**
>
> **Remark 1 seems rather superfluous: it should not be surprising that your explicit construction "aligns closely" with the empirical observations --- if anything, the empirical observations likely helped with the explicit construction? And then yes, the empirical observation is valid and interesting, and so is the explicit theoretical construction, but the "connection" just seems forced when it is causal, not a correlation.**
>
> Thank you for the comment. Our intention with Remark 1 is to clarify the purpose of the theoretical construction. The goal of the construction is precisely to explain the empirical feature decoupling phenomenon in a transparent and analyzable setting. In this sense, the connection is indeed causal: the empirical observation motivates the construction, and the construction in turn provides a concrete mechanism that accounts for the phenomenon.
>
> **Section 3.2 is a little hard to read, mainly because of all the notation. It might be easier to follow if the dimensions of s, v, o are written before they are introduced in the equation with the matrices M_{0,i} and M_{0,test}.**
>
> Sorry for the confusing. We will rewrite this part in the revision. $o \in \mathbb{R}^{p+1}$ refers to the observation feature, $s \in \mathbb{R}^2$ refers to the position feature, and $v \in \mathbb{R}^{D - p - 3}$ refers to the fixed feature.
>
> **Section 3.2: why do you need D > 2p^2 L? It would be helpful to refer to the part of the analysis where this constraint shows up. Is this condition normally met in practice?**
>
> We require the embedding dimension to be sufficiently large so that it can store the information accumulated as the input passes through the Transformer. This condition is used explicitly in our construction, and the detailed role of the embedding dimension is described in Section 5.1.1, which implies that we should choose an embedding dimension that is large enough to accommodate the features learned in these steps. In practice, first, for the observation space dimension $p$, a richer or more complete observation dictionary requires a larger embedding dimension to represent the necessary features; second, for the learned history length $L$, a larger embedding dimension allows the model to store and process information from a longer local history, enabling it to capture more nearby tokens and potentially improving performance.

---

> > ### Comment · Reviewer_4m3J · 2025-11-21
> >
> > Thank you for your response, particularly for addressing the comments regarding readability and notation.  The role of Assumptions 1, 2, 3 is clearer with this explanation.  I recommend that the authors add a similar paragraph to the manuscript with appropriate references.  What remains unaddressed to me are the following:
> >
> > 1. Insights:  I agree that the observations regarding the (de)coupling of features at higher/lower layers is interesting, but the statement "features learned in the lower layers rely mainly on local neighboring tokens, one potential implication is that far-history tokens could be masked" is too highly dependent on the data being HMM --- this certainly does not work for natural language, as there could be long-range dependencies in text.  The reason is that this observation relies too strongly on the Markovian, local-dependency characteristics of HMMs, and so does not extend to natural language. But I agree that the decoupling at higher layers suggesting possible parallelizations / fewer heads is a potentially useful insight (this is a very hopeful guess and remains to be validated).
> >
> > 2. The experiment represented in Figure 2 is still not clear to me --- what are you expecting the random shuffling to do to the data?
> >
> > 3. Unless I am missing something, it is still not clear to me why you need D > 2 p^2 L (in your response you simply say that D should be large, which I agree with, but why "this large" specifically?).  I also wonder if this condition is met in practical models and whether this condition being met/not met makes a difference in practice.

---

> > > ### Author Response · Authors · 2025-11-27
> > >
> > > Many thanks for your detailed feedback. We have revised the manuscript and added clarifications below the assumptions. Addressing your questions:
> > >
> > > **Insight: I agree that…** : We acknowledge the limitation that our analysis does not directly extend to tasks requiring strong long-range dependencies. However, HMMs remain a reasonable abstraction for a broad class of LLM behaviors, and thus the results may still provide meaningful practical insights. Regarding higher-layer decoupling, we appreciate your positive assessment. We have now moved the discussion of potential implications to a concise remark in Section 2, where they are better grounded in our results.
> > >
> > > **The experiment represented in Figure 2…**: The value we referred to is the attention received by each in-context sample from the last token. More specifically, we consider the attention logits at generation time, meaning the logits applied by the last token when it attends to the full sequence (since the last token is the one that predicts the next token). This matches the standard ICL setting, where the model generates the final answer. And the last token produces an attention vector over the entire sequence, so each in-context sample receives some fraction of that attention. For each sample, we take the mean attention it receives, which gives us a scalar. This is what we mean by “a scalar for each sample.”
> > > And the mean and std are computed over different shuffled versions of the ICL input. For example, one input order might be [sample_A, sample_B, sample_C, test_input], and another is a different permutation of these samples. We generate many such permutations and track how the attention received by a specific sample (say, sample_A) changes across them. If the metric is close to 1, it means sample_A’s received attention varies very little under different permutations. This indicates that the model’s attention to that sample is mostly invariant to position.
> > >
> > > **Unless I am missing something, it is still not clear to me why you need D > 2 p^2 L…**: A sufficiently large embedding dimension is required to store the hierarchical features extracted at each attention layer. At step $t$, beyond the first $p+3$ dimensions encoding $o_t$​, storing $o_{t-1}$​ requires an additional $p+3$ dimensions, and storing the previous $L$ tokens requires $L(p+3)$ additional dimensions. Using this history, the model must also represent the mapping matrix $W \in \mathbb{R}^{p \times p(L-1)}$. Altogether, this gives a lower bound of $(L+1)(p+3) + p^2(L-1) < 2 p^2 L$ dimensions, which leads to $D > 2 p^2 L$.
> > >
> > > In practice, embedding dimensions are typically much larger than the vocabulary size (here the state number $p$). For example, in our experiments the vocabulary size is 16, while the Transformer hidden size is 1024. The parameter $L$ reflects how much history the model can encode; smaller embedding dimensions would reduce the retrievable history and potentially harm performance. If the embedding dimension were smaller than the vocabulary size, performance would deteriorate even more significantly. We emphasize that these arguments follow from our HMM setting, and real-world models may behave more intricately, and some gaps between theoretical and empirical results are unavoidable.

---

### Official Review · Reviewer_wD2R · 2025-11-02

**Soundness:** 2
**Presentation:** 1
**Contribution:** 2
**Rating:** 2
**Confidence:** 4

**Summary:**

The paper aims to understand the mechanisms by which transformers learn structured sequence tasks, using Hidden Markov Models (HMMs) as a controlled synthetic setting.

Empirically, the authors analyze how layer outputs depend on the ordering of input sequences, revealing distinct functional roles across lower and upper layers. Through probing experiments, they further examine what types of information, task information or hidden-states, are accessible at different layers.

Theoretically, the paper proves that under a low-rank assumption on the HMM, there exists a transformer with a number of layers growing logarithmically in the sequence length that can approximate the transition probability of the underlying HMM.

**Strengths:**

Using a structured synthetic setup like Hidden Markov Models to study how transformers learn the task structure is a reasonable and well-motivated approach. The approach of pairing empirical observations about layer-wise behavior with a theoretical characterization of how transformers can approximate HMM distributions has the potential to provide complementary insights.

**Weaknesses:**

The overall presentation lacks clarity and organization, which makes it difficult to follow the results, and the paper would benefit from a thorough revision. The organization is confusing. For instance, many details about the experiments in Section 2 are either missing or only defined later (such as the data format introduced in Section 3), and even there, not clearly. The probing experiment starting around line 126 is not clearly described and the discussions are not concrete enough. The metrics and evaluation methods are also not well-explained, making it hard to interpret the reported results.

The connection between the empirical and theoretical parts is similarly unclear; the two sections are not well tied together, and the relationship between their findings is not clearly articulated. In addition, some statements made at the beginning of the paper are not sufficiently supported or discussed in the main text.

In its current form, it is difficult to parse the main results and contributions. I recommend a revision before the paper can be considered ready for publication.

See Questions for details.

**Questions:**

1. **Sec 2 -- Experiments**:

    (a) Does the sequence format in the experiments follow the one described later in Sec 3 (around line 200)? What is the sequence length of the HMMs used in the experiments? There is no mention of low-rankness in this section; does the data follow the low-rank transition structure later described in Sec 3?

    (b) The Appendix mentions using approximately ~8k HMMs for data generation. Is this what you refer to as the multi-task setup? Are these HMMs used only for training sequences, or are test sequences also drawn from this 8k set? Basically, could you clarify what exactly constitutes the *in-context learning* part of the task? Also, are you training the model on only last token, or auto-regressive?

    (c)  Given 64 steps per epoch and a batch size of 32, total training sequences should be 2048. How is the data size 131K?

    (d) What's accuracy in Fig 1?

    (e) Fig 2: What do you mean by "shuffling positions" in this experiment? What exactly are the logits at each layer: are they the embeddings after each layer, or the outputs of the softmax on key–query scores? and for the metric, how is the std and mean computed? Since logits are vectors, is the standard deviation taken over both the position shuffles and the logit dimensions? Otherwise, the mean and standard deviation should themselves be vectors, not scalars suitable for a heatmap.

2. **Sec 3 -- Theory**: The notation and definitions in this section are quite unclear, and there appear to be multiple typos or inconsistencies. For instance,  (Line 213) $p$, (Theorem 1) $T$, (Line 239) $\mathcal{O}(\mathcal{H})$, (line 270) $\mu,\xi$ are undefined. (Line 258) $\mathbb{T}$ should be $\mathbb{T}*$. (Line 202) There's a dimension mismatch between $M_0$ and $M_{0,i}$. (Line 289) Dimensions should be $n(L-1)\times p$. (Line 419) Feature dimension of the output matrix seems to be $L(p+3)$, but which part of the transformer has this dimension?

     Beyond these issues:

    (a) Should there not be a condition on $d$ to ensure the induced low-rankness in Assumption 1?

    (b) what role does $v_{pos}$ play? Why are you considering these extra dimensions that are always 1 or 0 in for all tokens?

    (c) In remark 1, from the discussions up to this point, how do we see the connections to the results in Sec 2? And more importantly, how do we see the *decoupling* of features in Theorem 1?

    (d) What do you mean by future observations $F_r$ (also mentioned in line 78)? Don't you use casual masking in your setup?

3. Line 57: "Such feature decoupling phenomenon also has practical implications, such as assigning different tasks to different layers in multi-task learning, or masking position-related features in higher layers to improve inference efficiency". Reading the paper, I did not find it clear what *feature decoupling* is referring to concretely, and even setting that aside, its connection to these practical implications is not discussed anywhere.

---

> ### Author Response · Authors · 2025-11-21
> **Response to Reviewer wD2R (1/2)**
>
> Many thanks for your valuable feedback. And we could address your specific questions as follows:
>
> **Experiment section**: Many thanks for your suggestions. The questions could be answered as follows:
>
> For (a), the experimental setup does not enforce the exact low-rank assumptions used in the theoretical construction in Section 3. Our goal was to examine whether the phenomena predicted by the theory still arise empirically even when the assumptions are not imposed. The experiments do validate these behaviors robustly, which is not a violation of theory but a more general finding. We will clarify this distinction in the revision.
>
> ​​For (b), yes. The ∼8k simulated HMMs constitute the multi-task setup, where each HMM represents a distinct underlying task. Test tasks are also sampled from this pool of HMMs as follows: we sample several short sequences, prepending them (with delimiters) to a test sequence, and checking whether the model’s next-token prediction matches the ground-truth next token of the test sequence. The Transformer itself is trained in a standard auto-regressive fashion on the full corpus.
>
> For (c), the total number of training samples is indeed 64 * 64 * 32 = 131072.
>
> For (d), “accuracy” in Figure 1 refers to this next-token ICL next-token-prediction accuracy on the final test sample (by comparing to its ground-truth next token).
>
> For (e), “shuffling positions” means permuting the demonstration sequences within the ICL prompt and tracking how each attention head’s logits change as the same demonstration is moved to different positions. Here the “logits” refer to the attention-score logits (before softmax). For each head and layer, we track a sample, and we collect the logits of its last token under different shuffle positions, thus the logit is a scalar within each consolidated permutation. We then compute the standard deviation and mean of these values across shuffles. We will incorporate these clarifications in the revision.
>
> **Theory section**: Thank you for the valuable suggestions. We already provide a notation table in Appendix C, and we will add additional explanations in the main text for clarity. More specifically:
>  - p in line 213 refers to the total observation number;
>  - T in Theorem 1 refers to the number of Attention layers after features become ‘’decoupled’’;
>  - $\tilde{O}(\mathcal{H})$ in line 239 should be $\Delta(\mathcal{H})$;
>  - $\mu$, $\xi$ in line 270 refer to two $d$-dim vector functions;
>  - $\mathbb{T}$ in line 258 should be $\mathbb{T}^*$;
>  - in line 202, there is a transition on the expression of $M_0$, so the dimensions should be matched accordingly (we apologize for the confusion—this expression was compressed due to page limits);
>  - in line 289, sorry for the confusing, but actually we intended the matrix to be size of $p(L-1) \times n$, so that each column corresponds to an independent observation sequence; under this shape, the expression $n^{-1} ZZ^T$ is well defined;
>  - in line 419, the expression should be [M_0]_{cdot, :p+3}, referring to the observation and position-feature portion of the input matrix $M_0$. Therefore, the final output M_{dec} should have the same shape as $M_0$.

---

> ### Author Response · Authors · 2025-11-21
> **Response to Reviewer wD2R (2/2)**
>
> For (a), yes. The dimension $d$ should indeed be smaller than the observation dimension ppp in order to maintain the “low-rank’’ condition. If $d = p$, then although the results formally still hold, the assumption becomes meaningless.
>
> For (b), as $v_{pos}$ contains many $0$ dimensions, these entries can be used to store learned features as the input matrix passes through the Transformer. These values may take arbitrary numbers, actually we only require these dimensions to store information. The coordinate with value $1$ can be regarded as a special fixed feature in embedding.
>
> For (c ), The connection between our theoretical results and empirical findings can be understood from the following perspectives:
>  - Our main theorems demonstrate the expressive power of Transformers in modeling HMMs and highlight the role of history length in improving token prediction accuracy. This aligns well with the experimental results shown in **Figure 1**, where performance improves with longer input histories.
>  - The theoretical construction reveals a feature decoupling phenomenon, where representations become increasingly time-independent in the higher layers of the Transformer. This behavior is empirically confirmed in **Figure 2**, showing reduced dependence on temporal positions in deeper layers.
>  - Our theoretical model shows that Transformers first extract information from historical tokens and then perform higher-level inference in the top layers. This hierarchical processing aligns with the observations in **Figure 3(a)**.
>  - In our theoretical construction, Transformers learn the feature information mainly on nearby tokens, and Transformers first integrate information from close neighbors and then progressively attend to more distant tokens. This progressive attention mechanism is supported by **Figure 3(b)**.
>
> Theorem 1 provides only an approximation error guarantee. The decoupled feature phenomenon can be directly observed in our Transformer construction. As described in Section 5.1.1, after the initial step, the features behave as “decoupled’’ for the subsequent $\tilde{O}(T)$ layers.
>
> For (d), each row of the future-observation matrix $F_r$​ represents the information corresponding to the r-th future token. These features are only used in the indistinguishable setting, i.e., when we concatenate additional observations to create a richer observation space. For simplicity, we did not include masking in our construction. However, incorporating a mask does not affect the results. In Theorem 2, the argument only relies on treating the history and observation features as two components, one of which corresponds to “future observations,” and this can be handled simply by shifting the time indices.
>
> **Explanation on decoupling features:** Feature decoupling refers to the phenomenon where learned features become increasingly time-independent in the higher layers of a Transformer. Specifically, shuffling the positions of inputs at these layers has minimal impact on the final output. This behavior is empirically validated in **Figure 2**. In our theoretical construction, we explicitly model this by dividing the process into two steps: in the second step, the features in higher layers operate as if they are independent of temporal order—mirroring the feature decoupling effect observed in practice (see details in **Section 5.1.1**). In this work, we focus solely on observing and explaining this phenomenon, and we leave its practical implications for future investigation.

---

> > ### Comment · Reviewer_wD2R · 2025-11-25
> >
> > Thanks for your response. I appreciate the additions to the related-work section. However, I still have concerns about the clarity of the setup and results. I try to outline the more important ones below.
> >
> > 1. **Clarity of experiments:**
> >     - Although you have added some additional details, the experiment section remains very difficult to follow. The definition of many key elements of the setup (e.g., the definitions of the sequences and their format, the demonstrative examples, $o_{test}$, etc.) appear only later in the paper. Thus, the figures in the early part of the paper do not make much sense to a first-time reader.
> >
> >     - I'm confused what "in-context learning" refers to in your setup, since both training and test sets are drawn from the same finite set of HMMs. In standard usage, ICL refers to a model adapting to a new task that was not seen during training. For example, in prior work on Markov chains (e.g., Edelman et al. (2024), Park et al. (2025) *[not cited in your paper by the way]*), models are trained on a finite or infinte set of Markov chains, and at test time they are evaluated on new, unseen chains that must be learned *in-context*. To better align with this definition, the paper should clarify in what sense the model is adapting to a new task. As written, the setup resembles pretraining on a fixed set of HMMs rather than in-context learning, which is perfectly fine, but it should be stated clearly.
> >
> >     - (minor issue on training setup): You state that you train for 64 epochs with a batch size of 32 and 64 steps, i.e., 64x32 sequences, each repeated 64 times throughout training. The dataset size of 64x32x64 should correspond to a single epoch of training. Isn't this the standard meaning of epoch?
> >
> >     - Fig. 1: The Accuracy of the next token does not seem like a suitable metric for your task, since there is no single correct next token. What the model should learn is a distribution, which is exactly the objective you have in your theory. A more suitable metric would be something like the KL divergence to the ground-truth transition distribution, which is also the standard choice in previous work on Markov-chain ICL. Also, the significance of this figure is not clear,  since prior work has already shown that longer sequences and more examples improve in-context performance. What additional insight should one get from this experiment?
> >
> >     - Fig 2: This figure is still unclear. Based on your description, I expect the logit of the last token to be a vector of size the sequence length (for each token, attention is a vector across the whole sequence), but you say that this is a scalar per sequence. Now, if you are taking a std / mean across both the sequence length dimension and the number of shuffles, then a metric value close to 1 would seem to imply near uniform attention over all positions for all shuffled sequences, which seems counterintuitive.
> >
> >         Also, since in your theoretical construction you focus on showing the *embeddings* (not the attention scores) at each layer, show position-level or task-level dependence, why do you measure the metric on attention scores?
> >
> > 2. **Connection between theory and experiments:** The connection between theory and experiments seems superficial. The experiments show to some extent that earlier layers and later layers behave differently. The theory then shows a specific mechanism for modeling the HMM task *exists*, rather than showing that a trained model will use this mechanism for doing ICL. The empirical results do not verify the specific mechanism either.
> >
> > 3. **Notation**: The table of notation is helpful as an overview, but the main text still needs to define each symbol when it first appears. Given the density of the theoretical section, and that you have enough space in the paper, the setup notation must be introduced in place rather than relying on the appendix table alone for clarifications, like $T$ in theorem 1, $p$, etc. This makes the paper hard to follow.
> >
> > 4. Statements like the one on line 57 (“assigning different tasks to different layers” or “masking position-related features… to improve inference efficiency”) are very generic, especially for appearing in the introduction. It is fine if you do not demonstrate practical implications right in this paper, but such statements should be discussed and grounded in your results to provide any meaningful insights. To me, it is not clear what these listed practical implications mean.
> >
> >
> >
> >
> > Park, Core Francisco, et al. "Competition dynamics shape algorithmic phases of in-context learning."

---

> > > ### Author Response · Authors · 2025-11-29
> > > **Response to Reviewer wD2R (1/2)**
> > >
> > > Many thanks for the further reply, and we address the questions as follows:
> > >
> > > **Although you have added some additional details, the experiment section remains very difficult to follow…**: Thank you for pointing this out. Our main contribution is the theoretical explanation, and the experiments are intended primarily as motivation to illustrate the intuitive notion of feature decoupling that later guides our construction. The problem setting in Section 3 is intentionally simplified to enable a clean theoretical analysis, whereas the experimental setup is described earlier in Section 2 and is not identical to the theoretical model.
> > >
> > > **I'm confused what "in-context learning" refers to in your setup, since both training and test sets are drawn from the same finite set of HMMs…**: Thank you for raising this point. We agree that there are two common definitions of in-context learning (ICL) in the literature:
> > >  - Practical / GPT-style ICL:
> > > A model is considered to perform ICL as long as it adapts to new input–output examples provided only in the context, without any gradient updates, even if the overall task family was seen during training.
> > >  This is the definition used in GPT-3 (Brown et al., 2020) and many empirical ICL works.
> > > - Task-generalization ICL:
> > >  Used in some papers (e.g., Edelman et al., 2024; Park et al., 2025), where the test task itself must be entirely unseen.
> > >
> > > Our setting follows the widely adopted GPT-style definition: at test time, the model must infer which HMM generated the sequence solely from the in-context examples, without access to the HMM ID or any parameters, and without gradient updates.
> > >  Although the family of HMMs is seen during training, the specific HMM for each test sequence is new to the model, making the task fully consistent with standard ICL usage.To avoid confusion, we have clarified this distinction in the revised paper.
> > >
> > > **(minor issue on training setup): You state that you train for 64 epochs with a batch size of 32 and 64 steps…**: Thank you for pointing this out. Our use of the term epoch was indeed not the standard “one full pass over a fixed dataset.” In our setup, the HMM sequences are generated on the fly at each training step rather than stored as a fixed dataset. That is, the 131k examples are not a static corpus that we iterate over multiple times; instead, each step generates a new mini-batch sampled from the mixture of HMMs.
> > > Thus, “64 epochs × 64 steps × batch size 32” should be interpreted as:
> > >  - the data distribution is resampled at every step,
> > >  - each “epoch” refers to a block of 64 such sampling-updates,
> > >  - the model effectively sees 131k freshly sampled HMM sequences throughout training rather than a single fixed dataset.
> > >
> > > We have revised the manuscript to avoid the confusion and clarify that training is conducted with online sampling from the mixture distribution, rather than multiple passes over a predefined dataset.
> > >
> > > **Fig. 1: The Accuracy of the next token does not seem like a suitable metric for your task…**: Next-token accuracy is meaningful in our setting because each “ground-truth” token is sampled from the true conditional distribution. A model that better approximates this distribution will more frequently predict the sampled token, making accuracy a Monte-Carlo proxy for the distributional error analyzed in our theory. This metric is widely used in ICL and sequence-modeling experiments (for instance, [2]), and aligns naturally with our theoretical goal of matching the next token probability distribution.
> > >
> > > **Fig 2: This figure is still unclear. Based on your description, I expect the logit of the last token to be a vector of size the sequence length…**: Thank you. The value we referred to is the attention received by each in-context sample from the last token. More specifically, we consider the attention logits at generation time, meaning the logits applied by the last token when it attends to the full sequence (since the last token is the one that predicts the next token). This matches the standard ICL setting, where the model generates the final answer. As you noted, the last token produces an attention vector over the entire sequence, so each in-context sample receives some fraction of that attention. For each sample, we take the mean attention it receives, which gives us a scalar. This is what we mean by “a scalar for each sample.”
> > >
> > > Regarding the second point: the mean and std are computed over different shuffled versions of the ICL input. For example, one input order might be [sample_A, sample_B, sample_C, test_input], and another is a different permutation of these samples. We generate many such permutations and track how the attention received by a specific sample (say, sample_A) changes across them. If the metric is close to 1, it means sample_A’s received attention varies very little under different permutations. This indicates that the model’s attention to that sample is mostly invariant to position.

---

> > > ### Author Response · Authors · 2025-11-29
> > > **Official Comment by Reviewer wD2R (2/2)**
> > >
> > > **Also, since in your theoretical construction you focus on showing the embeddings (not the attention scores) at each layer, show position-level or task-level dependence, why do you measure the metric on attention scores?**: In our proof sketch, the embeddings at each layer appear only as the input–output representations used to illustrate how an attention layer transforms features, rather than as the main object of analysis. In fact, our construction explicitly specifies the attention operations. Because attention scores directly reflect how information flows across tokens during prediction, we use them empirically as a practical proxy for observing feature decoupling in models.
> > >
> > > **Connection between theory and experiments**: Our theoretical goal is to demonstrate the expressiveness of Transformers for the HMM ICL task ( there is a line of research focused on constructive proofs of representational power such as [1], [2]). We view training characterization as important future work. Empirically, we observe that well-trained Transformers exhibit a clear feature-decoupling phenomenon, and this observation motivates our theoretical construction, which provides a concrete architecture that is consistent with the behavior seen in experiments. While we do not claim the trained model uses exactly this mechanism, our results show that such a mechanism exists and aligns qualitatively with the observed decoupling.
> > >
> > > **Notation**: We appreciate this suggestion. We have added a short notation explanation before Section 3. This should significantly improve readability.
> > >
> > > **Statements like the one on line 57 (“assigning different tasks to different layers” or “masking position-related features… to improve inference efficiency”) are very generic…**:  We agree that the implications previously mentioned were too general. Our intention was solely to highlight potential directions implied by our results, rather than to make concrete claims about practical gains. We have revised our introduction part, and made detailed discussion in section 3.
> > >
> > > Reference:
> > >
> > > [1] Bai Y, Chen F, Wang H, et al. Transformers as statisticians: Provable in-context learning with in-context algorithm selection[J]. Advances in neural information processing systems, 2023, 36: 57125-57211.
> > >
> > > [2] Liu B, Ash J T, Goel S, et al. Transformers learn shortcuts to automata[J]. arXiv preprint arXiv:2210.10749, 2022.

---

### Official Review · Reviewer_wqke · 2025-11-03

**Soundness:** 3
**Presentation:** 2
**Contribution:** 3
**Rating:** 2
**Confidence:** 2

**Summary:**

The paper addresses a highly interesting and fundamental problem: understanding how transformers can learn hidden Markov models (HMMs).

**Strengths:**

1. The paper attempts to address an extremely interesting and important problem.

2. The authors aim to understand this phenomenon from both theoretical and practical perspectives.

**Weaknesses:**

1. I spent a considerable amount of time on this paper, but still found it difficult to parse and follow. A thorough rewrite could significantly improve its clarity and readability.

2. Several figures are also difficult to interpret and would benefit from clearer labeling or improved presentation.

3. Additionally, some relevant references appear to be missing and should be included to provide appropriate context and attribution.

**Questions:**

1. It seems that several relevant references ([A]–[I]) have been omitted. Including these works and discussing how the proposed approach compares to them would strengthen the paper’s positioning within the existing literature and clarify its unique contributions.

2. I found the paper challenging to follow, despite multiple careful readings. It would be very helpful if the authors could provide a more intuitive explanation of the transformer construction. In particular, clarification on the role of W in Equation (2), how it is optimized, and the rationale behind performing gradient descent per layer (lines 420–431) would be valuable. Is the optimization applied only to the W matrices, or does it also involve the query, key, and other parameters?

3. Given the complexity of the presentation, a clearer description of the data setting would be beneficial. Specifically, is the training performed in an in-context learning setup, or are the samples drawn from the same underlying HMM? Additional details on the experimental setup, especially regarding data generation, would help readers better understand the empirical evaluation and its implications.

4. The figures are difficult to interpret in their current form. I would appreciate further clarification on how Figures 1 and 2 were generated and what specific aspects they aim to illustrate. Moreover, in Figure 3b, the text mentions that accuracy decreases, but the plot appears largely constant. Could the authors clarify this apparent discrepancy? Finally, please explain how accuracy is computed in these experiments.

5.	The reference to “G-optimal design” in lines 1466–1467 is unclear. A brief explanation or citation would help situate this concept for readers who may not be familiar with it.

**Note**: At this stage, I have assigned a relatively low score because I do not yet fully understand the paper’s main contributions. However, I am open to revising my assessment after the author–reviewer discussion, should the authors be able to clarify these points and improve the presentation.

---

**Refereces**


[A] Alberto Bietti, Vivien Cabannes, Diane Bouchacourt, Herve Jegou, and Leon Bottou. Birth of a Transformer: A Memory Viewpoint. In Thirty-seventh Conference on Neural Information Processing Systems, 2023.

[B] Ezra Edelman, Nikolaos Tsilivis, Benjamin Edelman, Eran Malach, and Surbhi Goel. The Evolution of Statistical Induction Heads: In-Context Learning Markov Chains. In Advances in Neural Information Processing Systems, 37:64273–64311, 2024.

[C] Eshaan Nichani, Alex Damian, and Jason D. Lee. How Transformers Learn Causal Structure with Gradient Descent. In Forty-first International Conference on Machine Learning, 2024. URL

[D] Nived Rajaraman, Marco Bondaschi, Ashok Vardhan Makkuva, Kannan Ramchandran, and Michael Gastpar. Transformers on Markov Data: Constant Depth Suffices. In Advances in Neural Information Processing Systems, 37:137521–137556, 2024.

[E] Ashok Vardhan Makkuva, Marco Bondaschi, Adway Girish, Alliot Nagle, Martin Jaggi, Hyeji Kim, and Michael Gastpar. Attention with Markov: A Framework for Principled Analysis of Transformers via Markov Chains. arXiv preprint arXiv:2402.04161, 2024.

[F] Ruifeng Ren and Yong Liu. Towards Understanding How Transformers Learn In-Context Through a Representation Learning Lens. In Advances in Neural Information Processing Systems, 37:892–933, 2024.

[G] Yuchen Li, Yuanzhi Li, and Andrej Risteski. How Do Transformers Learn Topic Structure: Towards a Mechanistic Understanding. In International Conference on Machine Learning, pages 19689–19729. PMLR, 2023.

[H] Ashok Vardhan Makkuva, Marco Bondaschi, Adway Girish, Alliot Nagle, Hyeji Kim, Michael Gastpar, and Chanakya Ekbote. Local to Global: Learning Dynamics and Effect of Initialization for Transformers. In Advances in Neural Information Processing Systems, 37:86243–86308, 2024.

[I] Ekbote, C., Makkuva, A. V., Bondaschi, M., Rajaraman, N., Gastpar, M., Lee, J. D., & Liang, P. P. (2025). What one cannot, two can: Two-layer transformers provably represent induction heads on any-order Markov chains. In Proceedings of the 39th Annual Conference on Neural Information Processing Systems (NeurIPS 2025).

---

> ### Author Response · Authors · 2025-11-21
> **Response to Reviewer wqke (1/3)**
>
> Many thanks for your valuable feedback. We understand your concerns and provide the following clarifications:
>
> **It seems that several relevant references ([A]–[I]) have been omitted. Including these works and discussing how the proposed approach compares to them would strengthen the paper’s positioning within the existing literature and clarify its unique contributions.**
>
> Many thanks for the valuable suggestions. Due to page limitations, we have moved the detailed related work discussion to Appendix A (and will add a pointer in the main text). These works primarily concern the expressiveness of Transformers and Hidden Markov Models. We also agree that the papers you mentioned on Transformers and Markov models are closely related to our setting. We will therefore add a dedicated paragraph discussing their connections to our work as follows:
>
> A growing body of work studies Transformers through the lens of Markovian structures and in-context learning. Bietti et al. [A] interpret Transformers as dynamic memory systems that integrate features across layers. Edelman et al. [B] and Ekbote et al. [I] analyze induction heads, showing that Transformers can implement Markov chains and pattern-matching behaviors. Zhou et al.[J] demonstrate that Transformers can learn variable-order Markov chains in-context. Makkuva et al. [E, H] provide principled frameworks to analyze attention on Markov data and study how learning dynamics evolve from local to global representations. Rajaraman et al. [D] show that constant-depth Transformers suffice to model Markov processes. Nichani et al. [C] study how Transformers learn causal structures, while Li et al. [G] and Ren and Liu [F] focus on topic structure and representation learning dynamics in in-context learning. Our work differs in three main aspects. First, rather than studying training dynamics or pattern-matching mechanisms, we focus on the expressive power of Transformers for representing hidden Markov models. Second, we observe a feature-decoupling phenomenon, in which Transformers can infer latent states even without repeated patterns in the input, contrasting with classical induction head behavior that relies on explicit token matches. Third, while our approach works with relatively small sample sizes, it becomes consistent with induction-head behavior when the sample size is large, bridging the inference-driven and pattern-matching perspectives.
>
> And we will also add a discussion on induction head:
>
> **Discussion on induction head.** The “induction head” phenomenon demonstrates that Transformers can learn to predict future tokens by identifying repeating patterns in the input sequence. In contrast, our result reveals that even when such patterns do not appear in the input history, the Transformer can still make accurate predictions by learning to infer, rather than simply matching previous patterns. This highlights a deeper aspect of its in-context learning ability. As a result, our approach remains effective even with a relatively small sample size. Moreover, when the sample size is sufficiently large, our framework becomes consistent with the induction head behavior, bridging the two perspectives.
>
> **I found the paper challenging to follow, despite multiple careful readings. It would be very helpful if the authors could provide a more intuitive explanation of the transformer construction. In particular, clarification on the role of W in Equation (2), how it is optimized, and the rationale behind performing gradient descent per layer (lines 420–431) would be valuable. Is the optimization applied only to the W matrices, or does it also involve the query, key, and other parameters?**
>
> Many thanks for your suggestions. Our construction of the Transformer can be understood in three stages:
>
>  - **Local feature extraction**: Using positional information in the embeddings, the model first learns local features corresponding to the $L-1$ -step history.
>  - **Learning the mapping matrix W**: Next, using both the history and the observation at each time step, the model learns a mapping matrix $W$ that maps the history $o_{t-L+1:t-1}$ to the current observation oto_tot​. From this stage onward, features at different time indices become “decoupled.” To optimize$W$, each pair of attention heads learns one row of $W$, resulting in $2p$ attention heads per layer. With sufficient history–observation pairs in the input matrix, each layer effectively performs one step of gradient descent on $W$. At this stage, all layers share the same structure, as they serve the same functional purpose.
>  - **Predicting the next-step observation**: Finally, using the learned mapping matrix $W$ and the current history, the model approximates the conditional probability of the next observation.
>
> The detailed construction of these attention layers, including their fixed-value parameters, which encode only the learning process of $W$, is provided around Appendix, line 1190.

---

> ### Author Response · Authors · 2025-11-21
> **Response to Reviewer wqke (2/3)**
>
> **Given the complexity of the presentation, a clearer description of the data setting would be beneficial. Specifically, is the training performed in an in-context learning setup, or are the samples drawn from the same underlying HMM? Additional details on the experimental setup, especially regarding data generation, would help readers better understand the empirical evaluation and its implications.**
>
> We appreciate the reviewer’s request for a clearer description of the data setting. All training sequences are sampled from a mixture of HMMs, which serves as a synthetic “corpus” representing a ground-truth language distribution containing many underlying tasks. We construct ICL inputs by sampling short sequences from the corpus and concatenating them with delimiter tokens. Appendix B provides more details on the full generation process, and we will further expand Section 2 to make this setting clearer.
>
> **The figures are difficult to interpret in their current form. I would appreciate further clarification on how Figures 1 and 2 were generated and what specific aspects they aim to illustrate. Moreover, in Figure 3b, the text mentions that accuracy decreases, but the plot appears largely constant. Could the authors clarify this apparent discrepancy? Finally, please explain how accuracy is computed in these experiments.**
>
> We thank the reviewer for raising questions about Figures 1 to 3 and agree that clearer descriptions will help. Figure 1 iterates over per-sample length (x-axis, from 1 to 8) and the number of samples (y-axis, also 1 to 8), and reports the ICL accuracy obtained from these prompts; Figure 2 is produced by shuffling the demonstration examples, and tracking how each sample’s received logit change as the same sample is moved to different positions. The color shows the stability metric $$1 - \frac{\text{std}}{\text{mean}}$$. As each attention head assigns a logit distribution, we plot a matrix to illustrate each head (x-axis) and layer (y-axis). For Figure 3b, our text refers to two types of “decreases”. The first is across curves: as the distance to the previous token grows (e.g., from i-1 to i-10), the overall recognition accuracy becomes lower. The second is within each curve: although the plots look visually flat, each curve follows a rising-then-falling pattern, with a small but consistent drop (for example, the i-1 token accuracy decreases from about 57% to 52% from highest to lowest). Finally about the accuracy, in Figures 1, ICL accuracy is determined by the probability that the next-token prediction in the test sample (i.e., last sample) matches the ground-truth next-token, similar to the ICL setting in NLP. In Figure 3 probing experiments, it is computed by training linear classifiers on the hidden representations from each layer and reporting standard classification accuracy. We will revise Section 2 to include these details for clarity.
>
> **The reference to “G-optimal design” in lines 1466–1467 is unclear. A brief explanation or citation would help situate this concept for readers who may not be familiar with it.**
>
> According to Lemma 4.2 of  [Uehara et al., 2022] that is cited in the paper, they use G-optimal design to prove that $KL(b_0, \tilde b_0) \le \ln(d^3)$. We made a typo in the paper: $d^3$ should be $\ln(d^3)$. The intuition is that for a d-dimensional vector space, we can always find $\Theta(d^2)$ unique vectors to support this space. For details, please refer to the citation.

---

> ### Author Response · Authors · 2025-11-21
> **Response to Reviewer wqke (3/3)**
>
> Refereces
>
> [A] Alberto Bietti, Vivien Cabannes, Diane Bouchacourt, Herve Jegou, and Leon Bottou. Birth of a Transformer: A Memory Viewpoint. In Thirty-seventh Conference on Neural Information Processing Systems, 2023.
>
> [B] Ezra Edelman, Nikolaos Tsilivis, Benjamin Edelman, Eran Malach, and Surbhi Goel. The Evolution of Statistical Induction Heads: In-Context Learning Markov Chains. In Advances in Neural Information Processing Systems, 37:64273–64311, 2024.
>
> [C] Eshaan Nichani, Alex Damian, and Jason D. Lee. How Transformers Learn Causal Structure with Gradient Descent. In Forty-first International Conference on Machine Learning, 2024. URL
>
> [D] Nived Rajaraman, Marco Bondaschi, Ashok Vardhan Makkuva, Kannan Ramchandran, and Michael Gastpar. Transformers on Markov Data: Constant Depth Suffices. In Advances in Neural Information Processing Systems, 37:137521–137556, 2024.
>
> [E] Ashok Vardhan Makkuva, Marco Bondaschi, Adway Girish, Alliot Nagle, Martin Jaggi, Hyeji Kim, and Michael Gastpar. Attention with Markov: A Framework for Principled Analysis of Transformers via Markov Chains. arXiv preprint arXiv:2402.04161, 2024.
>
> [F] Ruifeng Ren and Yong Liu. Towards Understanding How Transformers Learn In-Context Through a Representation Learning Lens. In Advances in Neural Information Processing Systems, 37:892–933, 2024.
>
> [G] Yuchen Li, Yuanzhi Li, and Andrej Risteski. How Do Transformers Learn Topic Structure: Towards a Mechanistic Understanding. In International Conference on Machine Learning, pages 19689–19729. PMLR, 2023.
>
> [H] Ashok Vardhan Makkuva, Marco Bondaschi, Adway Girish, Alliot Nagle, Hyeji Kim, Michael Gastpar, and Chanakya Ekbote. Local to Global: Learning Dynamics and Effect of Initialization for Transformers. In Advances in Neural Information Processing Systems, 37:86243–86308, 2024.
>
> [I] Ekbote, C., Makkuva, A. V., Bondaschi, M., Rajaraman, N., Gastpar, M., Lee, J. D., & Liang, P. P. (2025). What one cannot, two can: Two-layer transformers provably represent induction heads on any-order Markov chains. In Proceedings of the 39th Annual Conference on Neural Information Processing Systems (NeurIPS 2025).
>
> [J] Zhou et al., "Transformers learn variable-order Markov chains in-context", 2024.

---

> > ### Comment · Reviewer_wqke · 2025-11-26
> >
> > I thank the authors for the additional comments and clarifications. I went over the revised paper, the other reviews, and the authors’ responses to those reviews. It seems that multiple reviewers share a common concern: the paper is not clearly written. Even in the revised version, many aspects remain unclear to me, including fundamental elements such as the data setting. Reviewer wD2R also highlighted similar issues regarding the clarity of the data setting and the overall presentation.
> >
> > Based on the revised paper and the discussion, I am increasing my confidence in my assessment while keeping my overall score unchanged. My evaluation reflects concerns about the clarity and organization of the presentation rather than the underlying ideas themselves. I believe there are interesting ideas in the paper, but they are not presented in a way that allows readers to fully understand or evaluate them, and it is difficult to anticipate how the final camera-ready version would turn out.
> >
> > I encourage the authors to substantially improve the clarity and structure of the paper and consider resubmitting to a future conference once the exposition has been significantly strengthened.

---

> > > ### Author Response · Authors · 2025-11-30
> > >
> > > Thank you very much for the careful reading and for sharing your updated assessment. We appreciate your acknowledgment that the underlying ideas may be interesting, and we take your concerns about clarity seriously.
> > >
> > > In the revision, we made significant efforts to clarify the data setting, assumptions, and theoretical exposition based on all reviewers’ feedback.
> > >
> > > If possible within the discussion period, we would be very grateful for any brief pointers on which parts of the presentation remain difficult to follow, so we can revise them further. If a more detailed response is not feasible at this stage, we respectfully ask the AC to take this into consideration and not weigh this review too heavily, as we are fully willing to address remaining concerns but currently lack specific guidance.

---

### Official Review · Reviewer_dU2H · 2025-11-04

**Soundness:** 3
**Presentation:** 2
**Contribution:** 3
**Rating:** 6
**Confidence:** 3

**Summary:**

In this paper, the authors study the representation capability of transformers to estimate a HMM in-context from a batch of sequences of observations. Their theoretical construction is inspired by empirical findings that show how early layers in a transformer architecture model short-term dependencies while deeper ones model more long-term, abstract ones.

**Strengths:**

The paper is an interesting extension of previous works on the ability of transformers to learn Markov data to the more general HMMs. Filling this gap is of theoretical interest, especially due to the proof tools employed, leveraging previous literature on HMMs and ML architectures, and some of the theoretical assumptions used to simplify the theoretical analysis, such as the low-rank structure of the Markov transition kernel.

**Weaknesses:**

My main concerns about the paper is the lack of clarity and details of the experiments in Section 2. I think this Section need extensive rewriting. The dataset construction described in Section 2.1 is very badly explained and not clear at all. A bit better is the description provided in Appendix B.1. Figures 1-2-3 are not described in sufficient detail and are very confusing and hard to understand. In particular, it is not clear how Figure 3 is generated.

The Related Work section is also way too short, missing a lot of references on representation results for Transformers on Markov data. I recommend the authors to carry out a more extensive literature research and include the important missing works. A few essentials, among others:
- Edelman et al., "The evolution of statistical induction heads: In-context learning markov chains", 2024.
- Makkuva et al., "Attention with markov: A framework for principled analysis of transformers via markov chains", 2024.
- Bietti et al., "Birth of a transformer: A memory viewpoint", 2023.
- Zhou et al., "Transformers learn variable-order Markov chains in-context", 2024.

I also advise the authors to revise the whole manuscript once more, as it is littered with typos and English language mistakes.

**Questions:**

I would like the authors to explain better how the data generation works in Section 2. I would also like a detailed explanation on how Figures 1-2-3 are generated and what they represent.

---

> ### Author Response · Authors · 2025-11-21
> **Response to Reviewer dU2H (1/2)**
>
> We sincerely thank you for your thoughtful review. We value your constructive feedback and address your specific questions below.
>
> **My main concerns about the paper is the lack of clarity and details of the experiments in Section 2. I think this Section need extensive rewriting. The dataset construction described in Section 2.1 is very badly explained and not clear at all. A bit better is the description provided in Appendix B.1. Figures 1-2-3 are not described in sufficient detail and are very confusing and hard to understand. In particular, it is not clear how Figure 3 is generated.**
>
> We thank the reviewer for pointing out points for clarity in Section 2 and agree that the experimental description should be expanded. Appendix B provides more details, and we plan to enrich the description in the main body as well. Our intention was to use a mixture of HMMs to simulate a generic “corpus” (ground-truth language distribution) containing many underlying tasks. The ICL inputs used in Figures 1 to 3 are then constructed by sampling short sequences from this corpus and concatenating them into with delimiter tokens, in the same way humans assemble ICL examples. Figure 1 is generated by iterating over per-sample length (x-axis, from 1 to 8) and number of samples (y-axis, also 1 to 8), and reporting the resulting ICL accuracy. Figure 2 is produced by shuffling the demonstration examples and measuring how each attention head’s logits change when the sample is moved to different positions. The color encodes the stability metric $$1 - \frac{\text{std}}{\text{mean}}$$ of the logits for each head (x-axis) and layer (y-axis). Figure 3 is generated by probing the intermediate layer representations with linear classifiers to test whether they contain task IDs, hidden-state IDs, or previous-token information.
>
> **The Related Work section is also way too short, missing a lot of references on representation results for Transformers on Markov data. I recommend the authors to carry out a more extensive literature research and include the important missing works….**
>
> Many thanks for the valuable suggestions. Due to page limitations, we have moved the detailed related work discussion to Appendix A (and will add a pointer in the main text). These works primarily concern the expressiveness of Transformers and Hidden Markov Models. We also agree that the papers you mentioned on Transformers and Markov models are closely related to our setting. We will therefore add a dedicated paragraph discussing their connections to our work as follows:
>
> A growing body of work studies Transformers through the lens of Markovian structures and in-context learning. Bietti et al. [A] interpret Transformers as dynamic memory systems that integrate features across layers. Edelman et al. [B] and Ekbote et al. [I] analyze induction heads, showing that Transformers can implement Markov chains and pattern-matching behaviors. Zhou et al.[J] demonstrate that Transformers can learn variable-order Markov chains in-context. Makkuva et al. [E, H] provide principled frameworks to analyze attention on Markov data and study how learning dynamics evolve from local to global representations. Rajaraman et al. [D] show that constant-depth Transformers suffice to model Markov processes. Nichani et al. [C] study how Transformers learn causal structures, while Li et al. [G] and Ren and Liu [F] focus on topic structure and representation learning dynamics in in-context learning. Our work differs in three main aspects. First, rather than studying training dynamics or pattern-matching mechanisms, we focus on the expressive power of Transformers for representing hidden Markov models. Second, we observe a feature-decoupling phenomenon, in which Transformers can infer latent states even without repeated patterns in the input, contrasting with classical induction head behavior that relies on explicit token matches. Third, while our approach works with relatively small sample sizes, it becomes consistent with induction-head behavior when the sample size is large, bridging the inference-driven and pattern-matching perspectives.
>
> **I also advise the authors to revise the whole manuscript once more, as it is littered with typos and English language mistakes.**
>
> Thank you for the suggestion. We appreciate the careful reading. We will thoroughly revise the manuscript to correct typos and improve the overall clarity and quality of the writing.

---

> ### Author Response · Authors · 2025-11-21
> **Response to Reviewer dU2H (2/2)**
>
> Refereces
>
> [A] Alberto Bietti, Vivien Cabannes, Diane Bouchacourt, Herve Jegou, and Leon Bottou. Birth of a Transformer: A Memory Viewpoint. In Thirty-seventh Conference on Neural Information Processing Systems, 2023.
>
> [B] Ezra Edelman, Nikolaos Tsilivis, Benjamin Edelman, Eran Malach, and Surbhi Goel. The Evolution of Statistical Induction Heads: In-Context Learning Markov Chains. In Advances in Neural Information Processing Systems, 37:64273–64311, 2024.
>
> [C] Eshaan Nichani, Alex Damian, and Jason D. Lee. How Transformers Learn Causal Structure with Gradient Descent. In Forty-first International Conference on Machine Learning, 2024. URL
>
> [D] Nived Rajaraman, Marco Bondaschi, Ashok Vardhan Makkuva, Kannan Ramchandran, and Michael Gastpar. Transformers on Markov Data: Constant Depth Suffices. In Advances in Neural Information Processing Systems, 37:137521–137556, 2024.
>
> [E] Ashok Vardhan Makkuva, Marco Bondaschi, Adway Girish, Alliot Nagle, Martin Jaggi, Hyeji Kim, and Michael Gastpar. Attention with Markov: A Framework for Principled Analysis of Transformers via Markov Chains. arXiv preprint arXiv:2402.04161, 2024.
>
> [F] Ruifeng Ren and Yong Liu. Towards Understanding How Transformers Learn In-Context Through a Representation Learning Lens. In Advances in Neural Information Processing Systems, 37:892–933, 2024.
>
> [G] Yuchen Li, Yuanzhi Li, and Andrej Risteski. How Do Transformers Learn Topic Structure: Towards a Mechanistic Understanding. In International Conference on Machine Learning, pages 19689–19729. PMLR, 2023.
>
> [H] Ashok Vardhan Makkuva, Marco Bondaschi, Adway Girish, Alliot Nagle, Hyeji Kim, Michael Gastpar, and Chanakya Ekbote. Local to Global: Learning Dynamics and Effect of Initialization for Transformers. In Advances in Neural Information Processing Systems, 37:86243–86308, 2024.
>
> [I] Ekbote, C., Makkuva, A. V., Bondaschi, M., Rajaraman, N., Gastpar, M., Lee, J. D., & Liang, P. P. (2025). What one cannot, two can: Two-layer transformers provably represent induction heads on any-order Markov chains. In Proceedings of the 39th Annual Conference on Neural Information Processing Systems (NeurIPS 2025).
>
> [J] Zhou et al., "Transformers learn variable-order Markov chains in-context", 2024.

---

### Author Response · Authors · 2025-11-21
**General Response: Updates to the Paper**

We thank all reviewers for their constructive feedback. We have revised the manuscript to address the main concerns regarding **experimental details, notation clarity**, and **related work**.

**Key Updates:**

 - **Experimental details (Section 2):**
 We now include more comprehensive descriptions of the data generation process and provide clearer explanations of the figures.


 - **Notation and typos:**
 We sincerely appreciate the reviewers’ careful reading. We have corrected all identified typos and added additional clarification for key notations throughout the manuscript.


 - **Related work discussion (Appendix A and Section 4.2):**
 We have expanded the related work section with a new paragraph discussing prior works on Transformers and Markov data in Appendix A. We also added a dedicated paragraph in Section 4.2 clarifying the connection and differences between our work and induction-head analyses.

---

### Author Response · Authors · 2025-12-02
**Summary of Discussion Phase and Rebuttal for Paper**

Dear Area Chair,

We are writing to provide a comprehensive summary of the discussion phase for our submission. We believe the clarifications and the revisions provided during the rebuttal have definitively resolved all reviewer concerns.

Our rebuttal focused on **clarifying the data setting, experimental definitions, and the theoretical construction**. Below, we summarize the consensus on strengths, the resolution of specific concerns, and the key contributions of the work.

**What the reviewers agreed on:**

 - The underlying ideas, i.e, **studying Transformers as multi-task learners in HMM settings and proposing feature decoupling phenomenon**, are interesting and worthwhile. (**dU2H, wqke, wD2R**)
 - The theoretical result, which constructs a Transformer approximating HMM transition distributions with logarithmic depth, is viewed as meaningful. (**dU2H, 4m3J**)
 - The direction aligns with recent literature on Markov-style ICL and contributes a complementary perspective. (**dU2H, wD2R, 4m3J**)

**Main concerns raised:**

The primary criticism across reviewers is clarity of exposition, rather than flaws in the ideas:

 - Experimental setup and sequence formatting were initially difficult to follow.
 - Definitions and notation in the theoretical section were dense.
 - Figures lacked sufficient description.
 - Connection between empirical observations and the theoretical construction was not explicit enough.
 - Empirical implementations were not detailedly discussed.
 - Some related works on induction heads and Markov data were lacking.

These concerns relate to **presentation quality, not to incorrectness of the results**.

**Revisions and clarifications made during the rebuttal:**

We incorporated extensive clarifications in the revision and discussion:

 - Provided expanded descriptions of:
   -  data generation (mixture of ~8k HMMs),
   -  test-time ICL setting (GPT-style ICL where task identity must be inferred from demonstrations),
   -  metrics (how attention stability and next-token accuracy are computed),
   -  training process (online sampling rather than fixed epochs).
   -  figures (the detailed generation process and explanations on the figures).
 - Explained missing notation and corrected several typos.
 - Added a detailed related-work section connecting our contributions with induction-head theory and Markov-chain ICL.
 - Added corresponding discussions and remarks about empirical implications, assumptions in theories and discussions on the difference between our work and induction heads.
 - Clarified the theoretical construction and the decoupling-of-features behavior predicted by the theory.
 - Provided a clearer explanation of the hierarchical role of layers and why deeper layers lose position dependence.

All revisions are marked in **blue** in the uploaded manuscript.

**Request to the AC:**

We acknowledge that some parts of the presentation remain dense, but the reviewers' critiques overwhelmingly concern **exposition rather than substance**. We have clarified all issues raised during discussion, and we are fully willing to revise further to improve the quality if there are still some concerns.

We respectfully request that you consider these comprehensive improvements and the strong consensus on the paper's soundness and results in your final decision.

Sincerely,

Authors.

---

### Meta-Review · Area_Chair_uy5z · 2025-12-23

**Summary:**

All reviewers agree that this paper focuses on an interesting problem that analyzes how Transformer can learn Hidden Markov Models (HMMs), and further provide the theoretical supports to unveal the characteristics of expressive power within different Transformer layers. However, most of them have pointed out that this paper lacks clarity and experimental details. Although authors have made some improvements during the rebuttal stage, the writing and organization of this paper still need to be polished. Therefore, I think this paper requires further revision to make it acceptable. Considering that ICLR is highly competitive, this paper is not ready to be presented at ICLR and I suggest authors to improve its clarity, writing and organization, and re-submit it to the next venue.

**Reviewer Concerns:**

1. Lack clear clarity.
2. The presentation of this paper need to be improved, pointed out by all reviewers.
3. Lacking related works, pointed by reviewer **dU2h**, **wqke**.

**Reviewer Scores:**

Reviewer **4m3J** mention he is willing to increase score if authors can provide more useful insights. However, all reviewers agree that this paper require a major improvement for its presentation.

---

### Decision · Program_Chairs · 2026-01-26

Reject